# Engineering NIR-sighted bacteria

**Stefanie SM Meier[1], Michael Hörzing[1], Cornelia Böhm[2,3], Emma LR Düthorn[1], Heikki Takala[2], René Uebe[4,5], Andreas Möglich[1,5,6]\***

[1]Department of Biochemistry, University of Bayreuth, Bayreuth, Germany; [2]Department of Biological and Environmental Science, Nanoscience Center, University of Jyvaskyla, Jyväskylä, Finland; [3]Institute of Biochemistry, Graz University of Technology, Graz, Austria; [4]Department of Microbiology, University of Bayreuth, Bayreuth, Germany; [5]Bayreuth Center for Biochemistry and Molecular Biology, Universität Bayreuth, Bayreuth, Germany; [6]North-Bavarian NMR Center, Universität Bayreuth, Bayreuth, Germany

## eLife Assessment

This **important** study establishes bathy phytochromes, a unique class of bacterial photoreceptors that respond to near-infrared light (NIR), as versatile tools for bacterial optogenetics. NIR light is a key control signal in optogenetics due to its deep tissue penetration and the ability to combine with existing red- and blue-light sensitive systems, but thus far, NIR-activated proteins have been poorly characterized. The strength of evidence is **convincing**, with comprehensive in vitro characterization, modular design strategies, and validation across different hosts, supporting the versatility and potential for these tools in biotechnological applications. This study should advance the fields of optogenetics and photobiology and inspire future work.

**\*For correspondence:** andreas.moeglich@uni-bayreuth. de

**Competing interest:** The authors declare that no competing interests exist.

**Sent for Review** 25 April 2025
**Preprint posted** 27 April 2025
**Reviewed preprint posted** 01 July 2025
**Reviewed preprint revised** 15 October 2025
**Version of Record published** 03 November 2025

**Abstract** Spatially and temporally orchestrated gene expression underpins organismal development, physiology, and adaptation. In bacteria, two-component systems (TCS) translate environmental cues into inducible expression outputs. Inducible expression also serves as a versatile instrument in both basic and applied science. Here, we harness the photosensors of rhizobial bathy-phytochromes to construct synthetic TCSs for stringent activation of gene expression by near-infrared (NIR) light in laboratory and probiotic *Escherichia coli* strains, and in *Agrobacterium tumefaciens*. Orthogonal TCSs afford the multiplexed expression control of several genes by NIR and visible light. Notwithstanding substantial photochemical activation of bathy-phytochromes by visible radiation, the NIR-light-responsive systems hardly responded to red light. Evidently, light signals can be processed by TCSs into highly nonlinear responses at the physiologically relevant level of gene expression. These fundamental aspects likely extend to naturally occurring TCSs. Depending on their photosensor traits and environmental conditions, bathy-phytochromes may thus either be NIR-specific or function as colorblind receptors of light vs. darkness.

## Introduction

Proportionate to its essential roles in organismal physiology, development, replication, and adaptation, gene expression is stringently and precisely regulated. Various cellular circuits integrate cognate input signals and, in turn, ramp up or down the expression of target genes. Molecular analyses of these circuits not only benefit fundamental research, but also, they underlie applications in biotechnology and synthetic biology for the precise control of gene expression. Two-component systems (TCS) are the predominant signal-transduction pathways by which bacteria and several other organisms process environmental stimuli and convert them into intracellular gene-expression responses

(*Buschiazzo and Trajtenberg, 2019*; *Parkinson and Kofoid, 1992*). In their canonical form, TCSs comprise as one component a membrane-spanning, homodimeric sensor histidine kinase (SHK) that relays signals to the cell interior. SHKs generally consist of N-terminal extracellular/periplasmic sensor and C-terminal cytosolic effector modules, the latter of which subdivide into the DHp (dimerization and phospho-accepting histidine) and CA (catalytic) domains. In a signal-dependent manner, SHKs undergo autophosphorylation at a conserved, eponymous histidine residue to enable subsequent phosphoryl-group transfer to an aspartic acid residue within the response regulator (RR), that is, the second TCS component. Once phosphorylated, the RR commonly dimerizes and is thereby rendered capable of sequence-specific DNA binding and transcription initiation at target promoters (*Gao et al., 2019*). As it is the overall levels of phosphorylated RR that govern the physiological output of the TCS, many SHKs are bifunctional and also exert phosphatase in addition to their kinase activities (*Möglich et al., 2009*; *Russo and Silhavy, 1993*). Exceedingly stringent and steep regulatory responses can be achieved by finely balancing the opposing RR phosphorylation and dephosphorylation reactions. Depending on the signal, one or the other of the two elementary activities prevails, and the SHK consequently acts as either net kinase or net phosphatase. SHKs are specific for their respective RRs, thus allowing the coexistence in a single cell of multiple TCSs with minimal crosstalk (*Laub and Goulian, 2007*).

Among the plethora of known TCSs, certain representatives respond to light as their cognate stimulus (*Möglich, 2019*; *Ohlendorf and Möglich, 2022*). Pertinent TCSs comprise soluble photoreceptor SHKs that reside in the cytosol and absorb light at specific wavelengths within the near-ultraviolet (UV) to near-infrared (NIR) range of the electromagnetic spectrum (*Krell et al., 2010*; *Möglich, 2019*). Beyond their pivotal physiological role, these TCSs are of prime interest in biotechnology as they unlock the control of gene expression by light, an approach belonging to optogenetics (*Deisseroth et al., 2006*). As an inducer of gene expression, light brings several benefits in that it affords superior spatial and temporal precision; can be applied, dosed, and withdrawn with ease and in automatable fashion; and penetrates living tissue to a wavelength-dependent extent (*Weissleder, 2001*). Several light-responsive TCSs have become available and support strong regulatory responses covering the electromagnetic spectrum from the near-UV to the NIR (*Lazar and Tabor, 2021*; *Ohlendorf and Möglich, 2022*). Apart from little-modified naturally occurring setups (*Tabor et al., 2011*), additional photoreceptor SHKs have been engineered by modular recombination of sensor and effector entities (*Levskaya et al., 2005*; *Möglich et al., 2009*; *Möglich et al., 2010*). As a case in point, the widely deployed pDusk and pDawn plasmids resort to the blue-light-inhibited SHK YF1 originating from the fusion of a light-oxygen-voltage (LOV) photosensor to the DHp/CA effector of the *Bradyrhizobium japonicum* FixL SHK (*Möglich et al., 2009*; *Ohlendorf et al., 2012*). Irrespective of the wealth of setups for the optical control of bacterial gene expression, there is a dearth of circuits affording stringent responses to NIR light (*Ong et al., 2018*).

We previously expanded the pDusk plasmid family by exchanging said LOV photosensor for photosensory core modules (PCM) of bacterial phytochromes (BphP) (*Meier et al., 2024a*; *Multamäki et al., 2022*). The resultant pREDusk systems respond to red rather than blue light, which facilitates multiplexing (*Meier et al., 2024b*; *Multamäki et al., 2022*) with scores of bacterial optogenetic circuits that mostly sense blue light (*Ohlendorf and Möglich, 2022*). As an added benefit, red light penetrates mammalian tissue to a higher extent than blue light owing to reduced scattering and absorption by cellular constituents, hemoglobin, and other pigments (*Ash et al., 2017*; *Weissleder, 2001*). BphPs react to red and NIR light via biliverdin (BV) chromophores covalently attached within their PCMs (*Davis et al., 1999*; *Hughes et al., 1997*; *Takala et al., 2020*). The linear tetrapyrrole BV traverses between the *Z* and *E* isomers of its C15=C16 double bond which underpin the red-absorbing Pr and the far-red-absorbing Pfr states of BphPs, respectively. Red and NIR light triggers the photochromic *Z/E* isomerization of BV and thereby drives the photoconversion in the Pr→Pfr and Pfr→Pr directions, respectively. In conventional BphPs, as in those underpinning the pREDusk systems, the Pr state is the thermodynamically more stable form prevailing in darkness. After red-light exposure, these BphPs revert to their Pr resting state thermally within the dark-recovery reaction, which is generally protracted and plays out over many hours to days. Conventional BphPs therefore effectively act as ratiometric red/NIR-light sensors. In common with many natural BphP-SHKs (*Huber et al., 2024*; *Karniol and Vierstra, 2003*; *Lamparter et al., 2002*; *Multamäki et al., 2021*), the engineered ones in the pREDusk circuits exhibit net kinase activity towards their RR within their Pr states and net

phosphatase activity within their Pfr states. In pREDusk, red light therefore downregulates target gene expression by several hundred-fold (*Meier et al., 2024a*; *Multamäki et al., 2022*). More recently, we inverted the light response of the underlying BphP-SHKs by changing the composition of the linkers through which their sensor and catalytic modules transduce signals (*Meier et al., 2024a*). Doing so yielded BphP-SHKs with net kinase activity in their Pfr states and net phosphatase activity in their Pr states, thus supporting pronounced increases of gene expression under red light in the resultant pDERusk systems.

A second group of bacterial phytochromes, denoted bathy-BphPs, adopts the Pfr rather than the Pr state in darkness (*Karniol and Vierstra, 2003*). Consequently, NIR light promotes the Pfr→Pr conversion, as does visible light, if to a lesser degree. To the extent it has been studied, the thermal dark recovery in bathy-BphPs is generally fast and complete within seconds to minutes. Due to these aspects, bathy-BphPs are considered to primarily serve as sensors of light vs. darkness rather than of a specific light color (*Huber et al., 2024*).

Informed by spectroscopic analyses, we here harness the PCMs of select bathy-BphPs for stringently controlling TCS output and bacterial gene expression by NIR light. The resultant plasmid systems, dubbed pNIRusk, prompt stringent upregulation of gene expression under NIR light. Equally advantageously and unexpectedly, the engineered pNIRusk circuits show only minimal activation by red light. To enable multiplexed applications, we also advance light-regulated setups based on orthogonal TCSs. The pNIRusk plasmids not only afford pronounced NIR-light responses in *Escherichia coli* laboratory strains but also in the probiotic Nissle bacteria and in *Agrobacterium tumefaciens*. Taken together, we deliver versatile implements for the precision control of gene expression by NIR light in both fundamental and applied research.

## Results
### Bathy-bacteriophytochromes as sensors of light and pH

To subject bacterial gene expression to NIR light, we considered the PCMs of previously reported bathy-BphPs as input modules (*Rottwinkel et al., 2010*; *Tasler et al., 2005*). We better informed our choice by expressing, purifying, and spectroscopically characterizing BphP PCMs from *Azorhizobium caulinodans* (*Ac*PCM), *Agrobacterium vitis* (*Av*PCM), and *Pseudomonas aeruginosa* (*Pa*PCM). Consistent with earlier reports (*Rottwinkel et al., 2010*; *Tasler et al., 2005*), all three PCMs fully populated the Pfr state in darkness with distinct Soret- and Q-band absorbance peaks at 415 nm and (755±3) nm, respectively (*Figure 1a–c*). Exposure to NIR light drove the complete conversion to the Pr state characterized by maxima of the Soret band at (393±2) nm and of the Q band at around (698±3) nm. Illumination with blue and red light prompted partial population of the Pr state to around 65–80% and 40–70%, respectively. Strikingly, at pH 8.0, the Q-band intensity of the pure Pr state differed markedly across the PCMs, with *Pa*PCM showing the strongest signal, and *Ac*PCM the weakest. Based on earlier reports, these data may principally reflect differences in BV protonation at the pyrrole C ring (*Borucki et al., 2005*; *Zienicke et al., 2013*). To put this notion to the test, we recorded absorbance spectra for the three PCMs at pH values between 6 and 11 (*Figure 1d–f*). While the Q-band intensity remained invariant within the Pfr state, in the Pr state it increased with decreasing pH. By evaluating the corresponding absorbance peak heights as a function of pH, we determined apparent $pK_a$ values of 7.2±0.1, 8.0±0.1, and 9.9±0.2 for *Ac*PCM, *Av*PCM, and *Pa*PCM, respectively (*Figure 1g*). Following previous studies (*Zienicke et al., 2013*), we tentatively ascribe these $pK_a$ values to (de-)protonation of the nitrogen atom of the BV pyrrole C ring within the Pr state. In addition, pH also affects the photostationary Pr:Pfr ratio under red light which increases with rising alkalinity (*Figure 1h*, *Figure 1— figure supplement 1a-c*).

Having ascertained the bathy-character of the candidate PCMs and their Pr⇄Pfr interconversion, we next assessed the kinetics of reversion to their dark-adapted Pfr state at pH 8.0 following NIR illumination (*Figure 1i*). All three PCMs fully recovered to the Pfr state in single-exponential fashion, but the time constants $\tau$ for *Ac*PCM and *Av*PCM of (140±1) s and (80±1) s were markedly lower than the value of (610±1) s for *Pa*PCM. Closely similar recovery time constants resulted after exposing the samples to red instead of NIR light (*Figure 1—figure supplement 1d and e*). With rising alkalinity, we observed slower and less complete recovery after NIR-light exposure (*Figure 1—figure supplement 1g–l*). Potentially, this deceleration directly owes to BV deprotonation evidenced in the absorbance

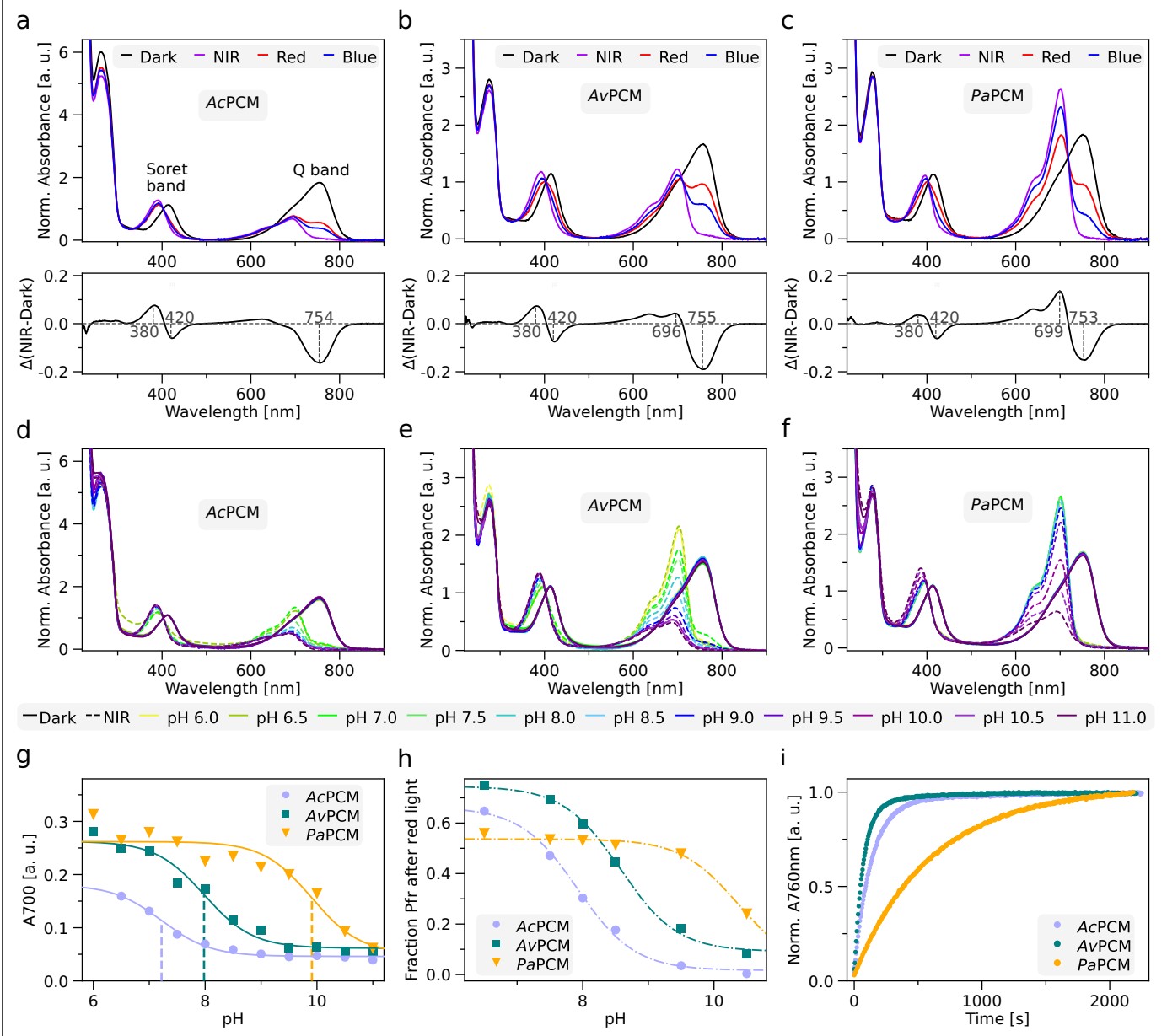

**Figure 1.** Spectroscopic characterization of bacteriophytochrome photosensory core modules (PCM) from *Azorhizobium caulinodans* (*Ac*PCM), *Agrobacterium vitis* (*Av*PCM), and *Pseudomonas aeruginosa* (*Pa*PCM). (**a**) UV/vis absorbance spectra of *Ac*PCM at pH 8.0 in darkness (black line), and after illumination with near-infrared (NIR, purple line), red (red), and blue light (blue). Data are normalized by the absorbance at 403 nm within the Soret band. The lower panel shows the NIR-dark difference absorbance spectrum with maxima and minima marked. (**b** and **c**) As in (**a**) but for *Av*PCM and *Pa*PCM. (**d**) Absorbance spectra of *Ac*PCM in darkness (solid lines) and after NIR illumination (dashed lines) at different pH values as indicated by color. (**e** and **f**) As in (**d**) but for *Av*PCM and *Pa*PCM. (**g**) Absorbance at 700 nm for *Ac*PCM (light purple circles), *Av*PCM (dark green squares), and *Pa*PCM (orange triangles) upon NIR-light exposure at different pH values. Data were fitted to the Henderson–Hasselbalch equation to determine apparent $pK_a$ values (dashed lines, *Ac*PCM: 7.2 ± 0.1, *Av*PCM: 8.0 ± 0.1, *Pa*PCM: 9.9 ± 0.2). (**h**) Calculated Pfr fractions of *Ac*PCM (light purple circles), *Av*PCM (dark green square), and *Pa*PCM (orange triangles) after red-light exposure at different pH values. The underlying absorbance spectra are shown in *Figure 1—figure supplement 1a–c*. (**i**) Dark-recovery kinetics at 25°C and pH 8.0 after NIR-illumination as followed by absorbance at 760 nm. Time constants for *Ac*PCM (light purple), *Av*PCM (dark green), and *Pa*PCM (orange) were determined by single-exponential fits.

The online version of this article includes the following figure supplement(s) for figure 1:

**Figure supplement 1.** Additional spectroscopic characterization of bacteriophytochrome photosensory core modules (PCM) from *Azorhizobium caulinodans* (*Ac*PCM), *Agrobacterium vitis* (*Av*PCM), and *Pseudomonas aeruginosa* (*Pa*PCM).

spectra. That is because BV remains protonated in the Pfr state across all investigated pH values, but within the Pr state it is progressively deprotonated at higher pH values. The return to the Pfr state hence requires re-protonation at elevated pH values, which may slow down the dark recovery.

## Subjecting bacterial gene expression to NIR-light control

To achieve control of bacterial expression by NIR light, we adapted the previous *Dm*REDusk plasmid which encodes a red-light-responsive chimeric SHK originating from the fusion of the PCM of the *Deinococcus maricopensis* BphP (*Dm*BphP) to the DHp/CA fragment of *B. japonicum* FixL (*Meier et al., 2024a*). As seen not least in the above spectroscopic experiments, bathy-BphPs also respond to visible besides NIR light, which stands to cause undesired crosstalk with genetic circuits that react to, for example, blue or red light. We reasoned that this adverse effect may be ameliorated by deliberately rendering the NIR-responsive systems comparatively insensitive to light. Guided by the above spectroscopic analyses, we selected *Ac*PCM and *Av*PCM as they exhibit a faster Pr→Pfr dark recovery than *Pa*PCM, which translates into a lower effective light sensitivity at photostationary state (*Ohlendorf and Möglich, 2022*; *Ziegler and Möglich, 2015*).

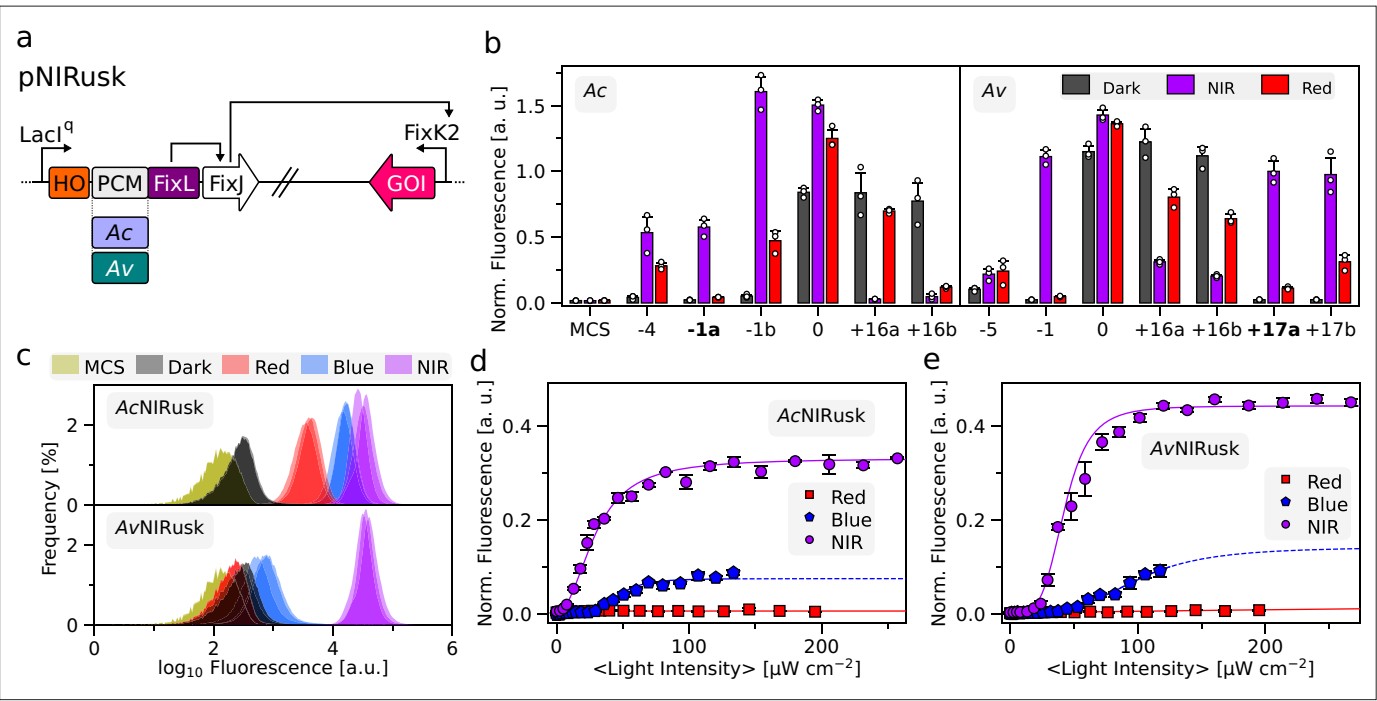

**Figure 2.** Engineering of light-regulated sensor histidine kinases (SHK) based on photosensory core modules (PCM) from bathy-bacteriophytochromes (BphP). (**a**) Schematic of the pNIRusk plasmids to control gene expression in bacteria by NIR light. The LacI$^q$ promoter constitutively expresses a tricistronic operon consisting of heme oxygenase (HO1), the light-regulated SHK (chimera of *Ac/Av*PCM and FixL), and the response regulator FixJ. When phosphorylated, FixJ binds to the FixK2 promoter and triggers the expression of a gene of interest (GOI). (**b**) Response of select *Ac*NIRusk (left) and *Av*NIRusk (right) systems to different light conditions (gray: darkness; purple: NIR light; red: red light). *Ds*Red reporter fluorescence was normalized by the optical density of the bacterial cultures. Individual pNIRusk variants are denoted by the relative lengths of the linkers within their underlying SHKs (see *Figure 2—figure supplement 2*), with the variants chosen for further analyses highlighted in bold. As control, an empty-vector construct (MCS) is shown, and data are normalized by the reference fluorescence value determined for the *Av*+17a variant. (**c**) Flow-cytometric analyses of bacteria containing *Ac*NIRusk-1a (top) or *Av*NIRusk+17a (bottom) upon incubation in darkness (black), blue (blue, 70 µW cm$^{-2}$), red (red, 100 µW cm$^{-2}$), or NIR light (purple, 200 µW cm$^{-2}$). An empty-vector control (MCS) is shown in green. (**d**) Light-dose response of bacteria containing *Ac*NIRusk-1a incubated in NIR (purple circles), red (red squares), and blue light (blue pentagons). (**e**) As in (**d**) but for *Av*NIRusk+17a. Data in panels b, d, and e represent mean ± s.d. of three biologically independent samples.

The online version of this article includes the following figure supplement(s) for figure 2:

**Figure supplement 1.** Multiple sequence alignment of the sensor histidine kinases within *Dm*REDusk, *Ac*NIRusk+0, and *Av*NIRusk+0.

**Figure supplement 2.** PATCHY library generation for *Ac*NIRusk and *Av*NIRusk.

**Figure supplement 3.** Setup for illumination with near-infrared (NIR) light at different intensities in a microtiter-plate (MTP) format.

**Figure supplement 4.** Additional characterization of the *Ac*NIRusk and *Av*NIRusk circuits.

With *Ds*Red as a fluorescence reporter gene, we replaced *Dm*PCM within *Dm*REDusk by either *Ac*PCM or *Av*PCM, while maintaining the same register for the linker between the PCM and DHp/CA moieties (*Figure 2—figure supplement 1*). In the resultant plasmids, referred to as pNIRusk, the heme oxygenase enzyme for intracellular BV provision, the BphP-SHK, and the matching RR FixJ thus form a constitutively expressed tricistronic operon (*Figure 2a*). The initial pNIRusk designs, denoted '0', showed considerable reporter fluorescence in darkness that only slightly elevated under both red and NIR light, despite their BphP-SHKs adhering to the same design as in *Dm*REDusk (*Figure 2b*).

Spurred by our recent construction of the pDERusk setups (*Meier et al., 2024a*), we hypothesized that light responsiveness may be installed by varying the composition of the linker between the PCM and DHp/CA moieties. That is because this linker is expected to adopt continuous α-helical coiled-coil conformation and serves as the conduit for conformational signals traveling from the PCM to the effector module (*Diensthuber et al., 2013*). Using the PATCHY approach (*Ohlendorf et al., 2016*), we generated libraries of the *Ac*NIRusk and *Av*NIRusk setups that differ in the length and sequence of said linker. In brief, the PATCHY protocol achieves linker variation by randomly recombining different portions of the linkers deriving from the parental receptors, that is, the bathy-BphP PCM and the FixL SHK, respectively (see *Figure 2—figure supplement 2*). By assessing the reporter fluorescence in darkness and under NIR and red light, we identified five light-switchable variants for *Ac*NIRusk and six for *Av*NIRusk (*Figure 2b*, *Figure 2—figure supplement 2*). Variants were considered light-switchable if they exhibited at least a twofold difference in reporter fluorescence in darkness vs. illumination with either red or NIR light. We refer to individual variants by their linker lengths relative to the initial constructs. Although the light-responsive variants all rely on the same PCMs and DHp/CA modules, they drastically differed in their response to NIR light, with some elevating and others lowering reporter-gene expression compared to darkness. Earlier application of PATCHY to SHKs responding to blue and red light (*Meier et al., 2024a*; *Ohlendorf et al., 2016*) had pinpointed sets of light-regulated receptor variants that exhibited systematic trends in the distribution of the underlying linkers. Specifically, receptor variants differing in their linker length by multiples of seven residues tended to show similar responses to light, which could be rationalized by the coiled-coil nature of said linkers (*Diensthuber et al., 2013*; *Ohlendorf et al., 2016*). At present, we only identified a handful of light-responsive SHK variants, thus largely precluding systematic analyses of the linker properties.

We subsequently focused on variants inducing gene expression under NIR light, albeit with intriguing variation in their response to red light. For instance, *Ac*NIRusk-1b with a linker one residue shorter than in *Dm*REDusk showed sizeable induction under red light. By contrast, the variant *Ac*NIRusk-1a with the same linker length but different sequence did much less so. For an eventual application, we deem a reduction of red-light-induced activation advantageous. We thus continued with *Ac*NIRusk-1a and *Av*NIRusk+17a, both of which strongly induce expression under NIR light compared to darkness but only show small responses to red light. For simplicity, we refer to these setups as *Ac*NIRusk and *Av*NIRusk in the following.

Flow cytometry revealed homogeneous single-cell fluorescence distributions for *E. coli* bacteria harboring *Ac*NIRusk or *Av*NIRusk (*Figure 2c*). When incubated in darkness, the median fluorescence was only twofold above the background fluorescence. Saturating NIR light (800 nm, 250 μW cm$^{-2}$) induced largely uniform shifts of the entire cell population to several 10-fold higher median fluorescence for *Ac*NIRusk and *Av*NIRusk, respectively, an effect similar in magnitude to the red-light response observed previously for the pDERusk systems (*Meier et al., 2024a*). At sub-saturating light doses, the single-cell fluorescence of bacteria containing *Ac*NIRusk and *Av*NIRusk uniformly rose to intermediate levels with increasing NIR-light intensity (*Figure 2—figure supplements 3 and 4a–b*). Red (100 μW cm$^{-2}$) and blue (70 μW cm$^{-2}$) light partially activated *Ac*NIRusk 11-fold and 35-fold, respectively (*Figure 2c*).

At the ensemble level, the average fluorescence per cell density ramped up sigmoidally with NIR light by up to around 70- and 150-fold for *Ac*NIRusk and *Av*NIRusk, respectively (*Figure 2d and e*). The two systems exhibited half-maximal activation at relatively high NIR-light intensities of (29±1) μW cm$^{-2}$ and (43±1) μW cm$^{-2}$, that is, about an order of magnitude above the half-maximal red-light doses previously determined for induction of the pREDusk and pDERusk systems (*Meier et al., 2024a*; *Multamäki et al., 2022*). Arguably, these differences reflect at least in part the respective dark-recovery times of the underlying bathy-BphPs which are in the seconds to minutes range rather

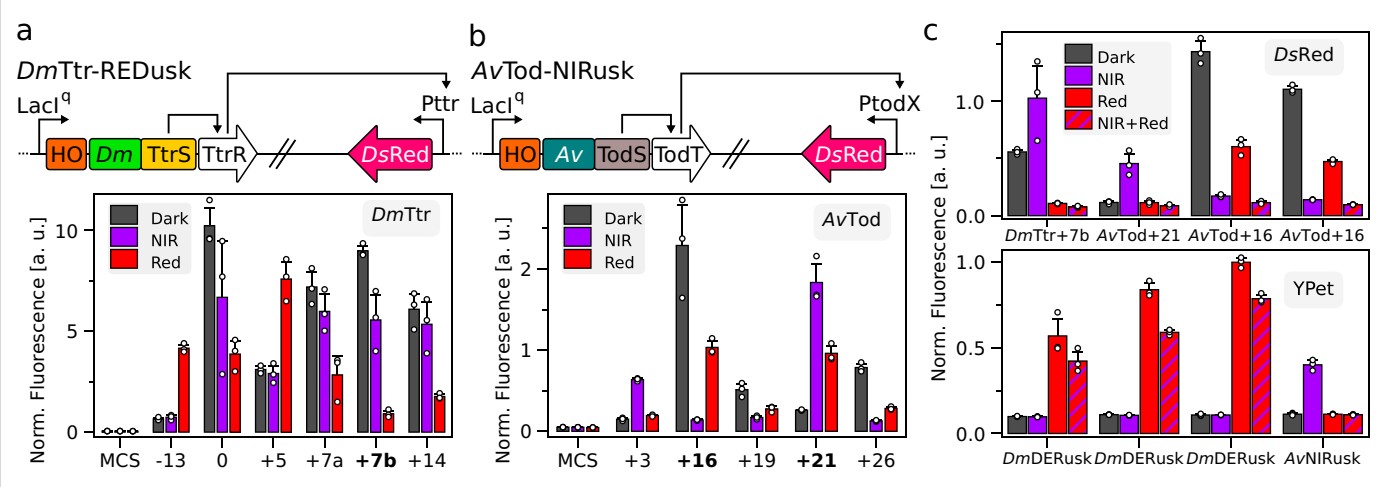

**Figure 3.** Engineering of orthogonal light-regulated two-component systems (TCS). (**a**) Schematic of the *Dm*Ttr-REDusk circuit based on the TtrSR TCS from *Shewanella baltica* combined with the photosensory core module (PCM) from the *Deinococcus maricopensis* bacteriophytochrome (BphP) (top). Reporter fluorescence of bacteria bearing circuits with different linker lengths of their underlying sensor histidine kinases (SHK) upon cultivation in darkness (gray), NIR (purple), and red light (red) (bottom). An empty-vector control is denoted as 'MCS', and data are normalized as in *Figure 2*. Variants are marked by the relative lengths of the linkers within their underlying SHKs compared to pREDusk (see *Figure 3—figure supplement 1*). Variants characterized further are highlighted in bold. (**b**) As in (**a**) but for the *Av*Tod-NIRusk circuit based on the TodST TCS from *Pseudomonas putida* combined with the bathy-BphP PCM from *Agrobacterium vitis*. (**c**) Multiplexing of bacteria containing two optogenetic circuits, as noted below the graphs, to separately control the expression of *Ds*Red (top) and YPet (bottom) reporters. After incubation in darkness (gray), NIR (purple), red (red), and combined NIR and red light (red and purple stripes), fluorescence was measured and, in the case of *Ds*Red, normalized as in *Figure 2*. YPet fluorescence was normalized by the reference value determined for *Dm*DERusk-YPet under red light (this panel). Data in panels a-c represent mean ± s.d. of three biologically independent samples.

The online version of this article includes the following figure supplement(s) for figure 3:

**Figure supplement 1.** PATCHY library generation for *Dm*Ttr-REDusk and *Av*Tod-NIRusk.

**Figure supplement 2.** Additional characterization of constructs used in multiplexing experiments.

than in the hours range for conventional BphPs. Bacterial cell growth was unaffected by NIR light (*Figure 2—figure supplement 4c–e*).

Red light also induced *Ac*NIRusk and *Av*NIRusk but only around two- and six-fold relative to darkness (*Figure 2d and e*). Blue light elicited a stronger 16- and 25-fold induction, respectively, at the highest probed light intensity of around 120 µW cm$^{-2}$. Higher blue-light intensities were not accessed owing to instrumental limitations and onsetting phototoxicity. From the data, we determined half-maximal blue-light doses of (46±3) µW cm$^{-2}$ and (100±20) µW cm$^{-2}$ for *Ac*NIRusk and *Av*NIRusk, respectively. Given the around 1.7-fold higher energy of blue compared to NIR light, the circuits therefore trigger at similar half-maximal photon fluencies for either light color. At least qualitatively, the degree of induction by the individual light colors (red, blue, NIR) follows the same trend as the photochemical responses evidenced in the underlying *Ac*PCM and *Av*PCM (see *Figure 1a–b*) where NIR light elicited the most pronounced Pr formation, followed by blue and red light.

## Engineering orthogonal light-responsive two-component systems

In common with earlier members of the pDusk pedigree, the pNIRusk systems are based on the FixLJ TCS from *B. japonicum* (*Anthamatten and Hennecke, 1991*). Combinations of several of these setups would hence experience crosstalk at the level of RR phosphorylation which, in all but a few cases, is undesired. To overcome this limitation, we set out to engineer light-responsive TCSs orthogonal to those used in pDusk, pNIRusk, and related setups. As recently surveyed, several TCSs have been characterized to the point that they have found application in synthetic biology (*Lazar and Tabor, 2021*). Based on their reported stringency of regulation, we selected as candidate TCSs TtrSR from *Shewanella baltica* that responds to tetrathionate and TodST from *Pseudomonas putida* that can detect toluene compounds (*Daeffler et al., 2017*; *Lau et al., 1997*).

Starting from *Dm*REDusk, we substituted the original FixL DHp/CA domains for those from TtrS, while also exchanging FixJ for the TtrR RR and putting the expression of a *Ds*Red reporter under control of the Pttr promoter that is subject to regulation by TtrSR (**Figure 3a**). Within this framework, we then varied the linker interconnecting the *Dm*PCM and the TtrS DHp/CA module via PATCHY (**Ohlendorf et al., 2016**). Screening of the resultant linker library for differential *Ds*Red expression under darkness, and NIR and red light identified several variants, termed *Dm*Ttr, with substantial light responses (**Figure 3a**, **Figure 3—figure supplement 1a**). Notably, the overall expression levels within the thus identified *Dm*Ttr systems were markedly higher than those in the pREDusk and pNIRusk setups.

Next, we extended the strategy to *Av*NIRusk by introducing TodST. Of note, TodS is a complex SHK with integrated phosphorelay that comprises in sequential order PAS-1, DHp-1, CA-1, REC, PAS-2, DHp-2, and CA-2 domains (**Busch et al., 2009**). Toluene signals, detected by PAS-1, modulate the activity of the DHp-1/CA-1 unit which phosphorylates the REC domain that in turn governs the activity of the DHp-2/CA-2 module and eventual phosphorylation of the TodT RR. Earlier reports showed that a miniaturized TodS variant only comprising the PAS-1, DHp-2, and CA-2 domains, dubbed Min-TodS, retains toluene sensitivity and stringent regulation of TodT (**Silva-Jiménez et al., 2012**). Consequently, we replaced the FixLJ TCS in *Av*NIRusk by the TodS DHp-2/CA-2 fragment, while also introducing the TodT RR and its cognate PtodX promoter (**Figure 3b**). Linker variation by PATCHY and screening for *Ds*Red reporter fluorescence pinpointed several variants, termed *Av*Tod, which either elevated or lowered gene expression in response to NIR light (**Figure 3b**, **Figure 3—figure supplement 1b**).

For further analyses, we selected the variants *Dm*Ttr+7b, *Av*Tod+16, and *Av*Tod+21 with PCM-DHp/CA linkers 7, 16, and 21 residues, respectively, longer than that in the parental *Dm*REDusk circuit. Exposed to red light, *Dm*Ttr+7b elicited an about 15-fold decrease of gene expression compared to darkness (**Figure 3—figure supplement 2a**). Whereas *Av*Tod+16 supported a 20-fold reporter-fluorescence decrease under NIR light, *Av*Tod+21 promoted an 8-fold increase of gene expression (**Figure 3—figure supplement 2b and c**). Although the attained regulatory efficiencies pale in comparison to those of *Dm*REDusk (**Meier et al., 2024a**) and *Av*NIRusk, the data show that the principal design and screening strategies readily translate to other TCSs.

To allow multiplexing of several light-responsive circuits in a single bacterial cell, we modified the *Dm*DERusk and *Av*NIRusk plasmids by replacing their kanamycin resistance marker, their ColE1 origin of replication (ori), and their *Ds*Red reporter gene by a streptomycin marker, the CloDF13 ori, and the yellow-fluorescent YPet reporter, respectively. The resultant *Dm*DERusk-Str-YPet and *Av*NIRusk-Str-YPet circuits showed similar responses to red and NIR light as the parental plasmids they derive from (**Figure 3—figure supplement 2d and e**). Pairs of compatible plasmids with different TCSs, oris, resistance markers, and fluorescent reporters were transformed into bacteria, followed by cultivation in darkness, under red (100 µW cm$^{-2}$) or NIR light (200 µW cm$^{-2}$), or under both light colors. By employing individual plasmid combinations, the expression of the *Ds*Red and YPet reporters could be individually controlled by the two light colors (**Figure 3c**, **Figure 3—figure supplement 2f**). For instance, when combining *Dm*DERusk-Str-YPet with *Av*Tod+21-*Ds*Red, YPet and *Ds*Red expression rose under red and NIR light, respectively, whereas the joint application of both light colors only induced the YPet reporter gene. Taken together, well-chosen plasmid sets enable the orthogonal expression control of two target genes by red and NIR light. Notably, the plasmids can be deployed as is with no further modifications required.

Although not probed here, the overall performance and relative response in multiplexed applications may be optimized by variation of the copy number and expression strength of the individual plasmids. Likewise, rather than using saturating light intensities as in the above experiments, yet finer-grained expression control could be exerted by adjusting the red- and NIR-light intensities. In a similar vein, different wavelengths, currently determined by the available instrumental setups, may be chosen to better separate the individual light responses of two (or, several) multiplexed systems.

## Applications in the laboratory and beyond

As TCSs widely recur across bacterial phyla, they often retain functionality upon transfer between different microbes. Earlier reports not least demonstrated that representatives of the pDusk family support light-regulated gene expression across several prokaryotes (**Ohlendorf and Möglich, 2022**). For instance, pDusk and its derivative pDawn (**Ohlendorf et al., 2012**) have been successfully deployed

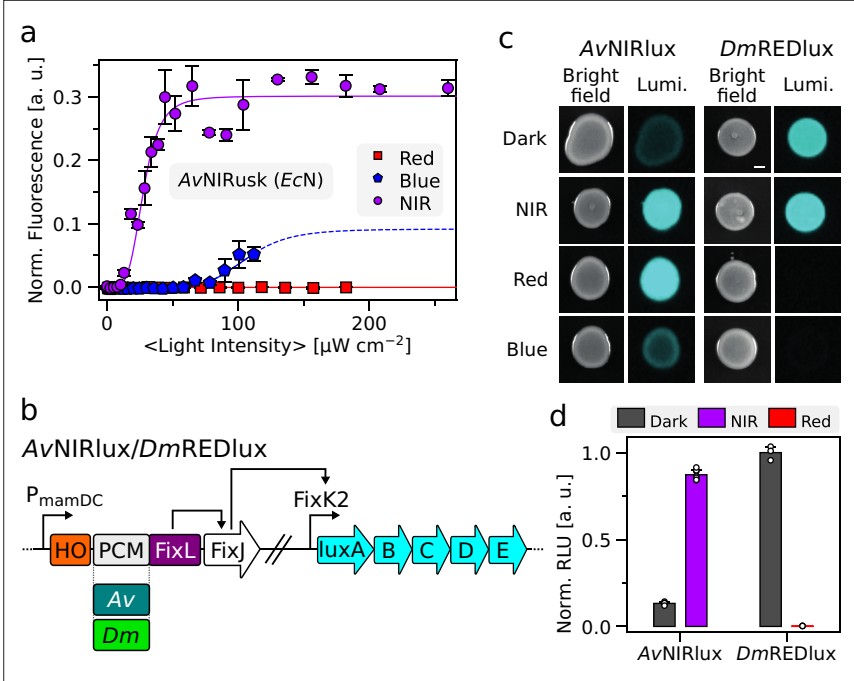

**Figure 4.** Light-regulated gene expression in *Escherichia coli* Nissle (*Ec*N) and *Agrobacterium tumefaciens.* (**a**) Light-dose response of *Ec*N harboring *Av*NIRusk after incubation at varying NIR- (purple circles), red- (red squares), and blue-light (blue pentagons) intensities. Corresponding data for *Ac*NIRusk are shown in *Figure 4—figure supplement 1*. (**b**) Schematic of the *Av*NIRlux and *Dm*REDlux circuits based on *Av*PCM and *Dm*PCM. The tricistronic cassette comprising heme oxygenase (HO1), the sensor histidine kinase (chimera of *Av/Dm*PCM and FixL), and FixJ is controlled by the constitutive PmamDC promoter from *Magnetospirillum gryphiswaldense*, whereas the FixK2 promoter regulates the expression of the *luxABCDE* operon. (**c**) *A. tumefaciens* bacteria bearing *Av*NIRlux (left) and *Dm*REDlux (right) were spotted on plates and incubated at different light conditions (darkness, NIR, red, or blue light), followed by luminescence (Lumi.) measurements (see *Figure 4—figure supplement 2a*). (**d**) *A. tumefaciens* bacteria carrying *Av*NIRlux and *Dm*REDlux were incubated in liquid culture for 20 h under different light conditions (gray: darkness, purple: NIR light, red: red light), followed by recording of reporter luminescence (see *Figure 4—figure supplement 2c and d*). Data in panels a and d represent mean ± s.d. of three biologically independent samples.

The online version of this article includes the following figure supplement(s) for figure 4:

**Figure supplement 1.** Gene expression regulated by *Ac*NIRusk in *Escherichia coli* Nissle.

**Figure supplement 2.** Gene expression regulated by *Av*NIRlux and *Dm*REDlux in *Agrobacterium tumefaciens.*

in *E. coli* Nissle 1917 (*Ec*N) and other probiotic bacteria (*Cui et al., 2021*; *Magaraci et al., 2014*; *Pan et al., 2021*; *Sankaran and del Campo, 2019*; *Yang et al., 2020*). Not least, these experiments augur the application of light-responsive bacteria in the gut of mammals as programmable, living therapeutics (*Ohlendorf and Möglich, 2022*). To date, however, pertinent applications are limited by the shallow tissue penetration of blue light required for triggering pDusk and pDawn. Although upconverting nanoparticles can be administered jointly with the light-responsive bacteria to allow actuation by NIR light, this approach introduces complexity and departs from a fully self-contained, optogenetic strategy (*Yang et al., 2020*). To assess whether the presently generated light-responsive gene-expression systems may overcome this potential limitation, we introduced *Ac*NIRusk and *Av*NIRusk into *Ec*N. Both setups mediated a more than 100-fold upregulation of reporter-gene expression under NIR light with half-maximal doses of (27±1) and (28±2) μW cm$^{-2}$, respectively (*Figure 4a*, *Figure 4—figure supplement 1*). Of advantage, the response to blue and red light was even lower than observed above for the *E. coli* laboratory strain.

We next probed the performance of *Av*NIRusk and *Dm*REDusk in the biotechnologically relevant *A. tumefaciens*. To this end, we constructed derivative plasmids, denoted *Av*NIRlux and *Dm*REDlux, that subject the expression of a luciferase operon under the control of the light-responsive TCSs

(*Figure 4b*). After conjugational transfection into *A. tumefaciens*, the light response was assessed by spotting the bacteria on agar plates (*Figure 4c*). As visualized by luminescence, the *Av*NIRlux system strongly induced expression upon incubation under NIR light compared to darkness. Different from *E. coli* (see *Figure 2e*, *Figure 4—figure supplement 2a*), red light activated to similar extents as NIR light, whereas blue light induced expression only weakly. In the case of *Dm*REDlux, the expression of the *lux* reporter was strongly reduced under both blue and red light, but not under NIR light or in darkness. Although *A. tumefaciens* harbors two endogenous BphPs (*Karniol and Vierstra, 2003*), our experiments did not detect any differences in growth or phenotype between cultivation under NIR and red light. We next characterized the response of *A. tumefaciens* cultured in liquid medium. Once reaching the stationary phase after around 20 h (*Figure 4—figure supplement 2b*), the cultures bearing *Av*NIRlux exhibited a sevenfold activation of *lux* expression by NIR light compared to darkness. For *Dm*REDlux, the *lux* expression was around 300-fold higher in darkness than under red light (*Figure 4d*, *Figure 4—figure supplement 2c-d*).

## Discussion

### Bathy-bacteriophytochromes as tunable sensors of light intensity and quality

As conventional bacteriophytochromes do, bathy-BphPs transition between their Pr and Pfr states with 15*Z*- and 15*E*-configured biliverdin, respectively. In marked contrast to the conventional representatives, bathy-BphPs assume their 15*E*-configured Pfr state as the thermodynamically most stable form in darkness, rather than their Pr state. Despite this principal difference, the overall PCM architecture and the residues lining the bilin binding pocket are remarkably conserved across known BphPs. Notwithstanding limited successes in reprogramming certain BphPs (*Böhm et al., 2025*; *Xu et al., 2024*), the molecular determinants underlying bathy-character remain elusive and their design demanding.

Owing to their Pfr dark-adapted state, bathy-BphPs undergo net Pfr→Pr conversion when irradiated with light within the near-UV to NIR region of the electromagnetic spectrum. Put another way, there is no light quality that would fully populate the dark-adapted Pfr state. Bathy-BphPs surmount this challenge by a fast thermal Pr→Pfr recovery that restores the Pfr state within seconds to minutes. The primary physiological role of bathy-BphPs may hence be that of sensors of light quantity rather than quality (i.e., color) (*Huber et al., 2024*).

Intriguingly, we uncovered a profound pH influence on photoreception in bathy-BphPs. With rising alkalinity, the absorption cross-section of the Q band strongly decreases within the Pr state but remains unchanged for Pfr. Based on earlier studies (*Borucki et al., 2005*; *Velazquez Escobar et al., 2015*; *Zienicke et al., 2013*), we attribute these spectral changes to (de-)protonation of the nitrogen atom of the BV pyrrole C within the Pr state. The pertinent $pK_a$ values range between 7–8 for *Ac*PCM and *Av*PCM, similar to the value of 7.6 for the bathy-BphP Agp2 from *A. tumefaciens* (*Zienicke et al., 2013*), but substantially lower than the value of about 10 for *Pa*PCM. Inspection of the experimentally determined *Pa*PCM and Agp2PCM structures in their Pfr states (*Schmidt et al., 2018*; *Yang et al., 2008*) and the corresponding Pfr models for *Ac*PCM and *Av*PCM (*Abramson et al., 2024*) reveals highly conserved chromophore-binding pockets with most residues adjacent to BV identical in all four proteins (*Figure 5—figure supplement 1*). However, *Pa*PCM features tyrosine and serine residues at positions 190 and 275 rather than phenylalanine and alanine, respectively, in the equivalent places of the other three PCMs. Although speculative at present, these systematic differences may render the *Pa*PCM binding pocket comparatively hydrophilic and thereby account for the divergent protonation equilibrium of the pyrrole C ring.

BV deprotonation within the Pr state at elevated pH incurs at least three principal effects that under constant light all promote population of the Pr state at the cost of the Pfr state. First, the Pr→Pfr dark recovery slows down between pH 7 and 10 for all three investigated PCMs. At least in part, this may reflect the need for BV re-protonation upon return from the Pr to the Pfr state which remains fully protonated at all investigated pH values. Second, as noted above, BV deprotonation leads to a drastic reduction of the Q-band absorbance within the Pr state (see *Figure 1*). For a given intensity of visible light, photoconversion to the Pfr state thus becomes less effective, thereby again favoring the Pr state at photostationary state. Third, BV deprotonation also goes along with a dramatic decrease

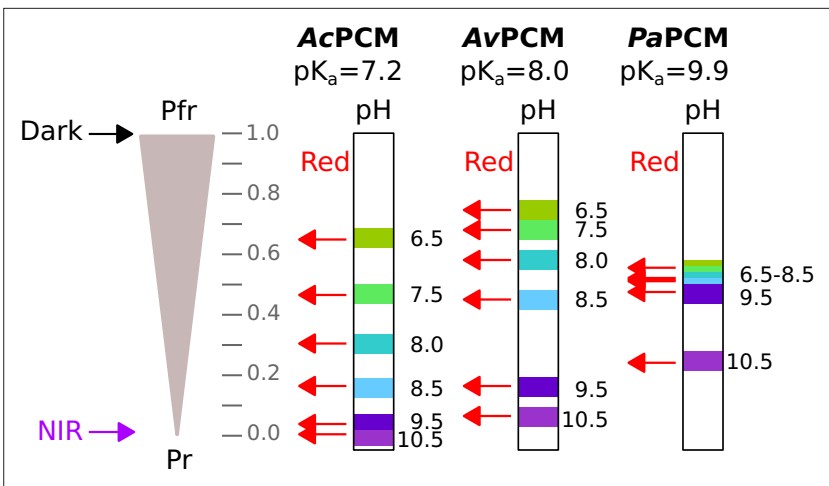

**Figure 5.** Photoreception by select bathy-bacteriophytochromes (BphP) at different pH values. Throughout the entire tested pH range, the photosensory core modules (PCM) of the bathy-BphPs from *Azorhizobium caulinodans* (*Ac*PCM), *Agrobacterium vitis* (*Av*PCM), and *Pseudomonas aeruginosa* (*Pa*PCM) adopt a pure Pfr state in darkness with fully protonated biliverdin (BV) chromophores. NIR light drives the complete conversion to the Pr state in which BV can be either protonated or deprotonated as governed by pH and the respective $pK_a$ value of the bathy-BphP. Red light populates a photostationary mixture with Pr and Pfr proportions strongly depending on pH. At high pH values, red light can drive the bathy-BphP nearly completely to the Pr state for *Ac*PCM and *Av*PCM. Owing to a higher $pK_a$ value for BV protonation, *Pa*PCM is less affected but follows the same trend.

The online version of this article includes the following figure supplement(s) for figure 5:

**Figure supplement 1.** Molecular environment of the bilin chromophore in bathy-bacteriophytochromes.

in the apparent photoconversion quantum yields in the Pr→Pfr vs. the Pfr→Pr directions (*Figure 1—figure supplement 1f*). As best exemplified for *Ac*PCM, at pH 6.5, the respective quantum yields for the photoreversible Pr⇄Pfr interconversion are about equal, but at pH 10.5, the quantum yield in the Pr→Pfr direction is less than 1% that in the Pfr→Pr direction. Together with the pH-dependent reduction in the Pr-state Q-band absorbance, this mechanism leads to vastly different red-light-induced photostationary states as a function of pH (see *Figure 1h*). Whereas at low pH, saturating red light results in about 60–70% Pfr population, at high pH values almost no Pfr state remains (*Figure 5*). As a direct consequence, at pH 10.5, red light counterintuitively drives the complete conversion to the Pr state of *Ac*PCM. Put another way, in its deprotonated form this bathy-BphP essentially becomes colorblind and acts as a pure light-intensity sensor. Although less pronounced, *Av*PCM and *Pa*PCM follow the same trend with pH.

While it is debatable whether the rhizobia *Ac*PCM and *Av*PCM derive from ever encounter such extreme pH swings, it has been proposed that the resultant changes in photochemical properties could serve a sensing function (*Zienicke et al., 2013*). Even independent of this aspect, our present analyses show how pH impacts bathy-BphP photoreception at multiple levels. While it remains true that all light colors net promote the Pfr→Pr transition, they do so to different extents which can have a profound impact on downstream signal transduction as discussed below. We propose that modulation of the BV protonation midpoint has been harnessed during evolution to sculpt the light response of extant bathy-BphPs according to the individual needs of the respective organism of origin. In line with this hypothesis, we presently evidence a considerable span of $pK_a$ values for BV protonation across several bathy-BphP PCMs.

## Engineering of NIR-light-responsive proteins

As best exemplified by sensor histidine kinases, PCMs of bathy-BphPs can occur in conjunction with the same type of effector as those of conventional BphPs (*Blum et al., 2025*). However, it has been unclear whether a single, specific output protein can be regulated in its activity by both PCM types. The successful recombination of bathy-BphP PCMs with the same DHp/CA module earlier controlled by conventional BphP PCMs (*Meier et al., 2024a*; *Multamäki et al., 2022*) now demonstrates the

principal validity of this idea. At least to some extent, the PCMs of conventional and bathy-BphPs are hence interchangeable, a finding that indicates shared or compatible signal-transduction mechanisms, as also suggested by the conserved structure and photochemistry across the BphP class (*Möglich, 2019*; *Takala et al., 2020*). As one important ramification, numerous previously engineered red-light-responsive receptors may thus be subjugated to NIR-light control by swapping out their underlying PCMs from conventional BphPs for ones from bathy-BphPs (*Chernov et al., 2017*; *Etzl et al., 2018*; *Leopold et al., 2022*; *Leopold et al., 2019*; *Levskaya et al., 2005*; *Schmidl et al., 2014*; *Stabel et al., 2019*; *Stüven et al., 2019*; *Tang et al., 2021*). Compared to the challenging approach of imparting bathy-character via limited PCM modifications (*Böhm et al., 2025*; *Xu et al., 2024*), the design task thus considerably simplifies.

We identify several caveats that likely hold for future design endeavors, too. Notably, the connection between PCM and effector of choice needs to be conducive to transmitting light signals. When at present employing the same linker register that supports red-light-inhibited gene expression in *Dm*RE-Dusk, we initially derived *Ac*PCM- and *Av*PCM-based systems giving rise to largely constitutive target gene expression without pronounced light responses. Adjusting the length and sequence of the linker intervening the PCM and the DHp/CA domains then yielded variants with robust gene-expression responses to NIR light. Based on structural and biochemical evidence, said linker is expected to form a continuous parallel α-helical coiled coil within the homodimeric SHK (*Bódizs et al., 2024*; *Burgie et al., 2024*; *Diensthuber et al., 2013*; *Jacob-Dubuisson et al., 2018*; *Malla et al., 2024*; *Möglich, 2019*; *Möglich et al., 2009*). Although other BphP-based photoreceptor designs may involve linkers adopting different structures, the variation and probing of length and sequence variants generally represents a worthwhile strategy that may eventually bear fruit. As, depending on the use case, size-able variant libraries may have to be assessed, two principal aspects emerge that benefit the implementation of this strategy. First, the construction of libraries covering the desired diversity in linker composition should be efficient, which can, for example, be achieved by resorting to the PATCHY protocol (*Ohlendorf et al., 2016*) as done here. Second, the screening for light-regulated function should afford sufficient throughput, for example, by coupling photoreceptor function to a fluorescent readout or cell survival.

## Tunability of NIR-light-activated two-component systems

The recombination of bathy-BphP PCMs with a SHK effector module presently led to the pNIRusk plasmids for stringent activation by NIR light of bacterial gene expression. As the underlying PCMs absorb light over the entire near-UV to NIR region of the electromagnetic spectrum, the pNIRusk systems are expected to also activate when exposed to other light colors. To dampen the commonly undesirable activation by visible light and thereby facilitate the multiplexing with other photoresponsive circuits, we deliberately chose *Ac*PCM and *Av*PCM because of their comparatively fast Pr→Pfr thermal recovery. At photostationary state under constant illumination, a faster recovery results in an effective desensitization against light, be it of blue, red, or NIR color (*Ziegler and Möglich, 2015*). Consistent with this design rationale, the *Ac*NIRusk and *Av*NIRusk setups indeed exhibit half-maximal light doses around ten times higher than for, for example, *Dm*DERusk which relies on a conventional BphP PCM (*Meier et al., 2024a*). Given its low phototoxicity, NIR light may be applied at even high powers without harming the biological system under study, thus enabling the full activation of the pNIRusk systems irrespective of their fully intended relatively low light sensitivity. We note that NIR light around 800 nm is commonly used in a therapeutic setting for photobiomodulation at intensities exceeding the ones presently used by up to several orders of magnitude (*Hamblin, 2016*).

Perplexingly, the *Ac*NIRusk and *Av*NIRusk circuits respond moderately or very weakly to blue and red light, respectively (see *Figure 2d and e*). Although the strength of the gene-expression response monotonically increases with the extent of Pfr→Pr photoconversion elicited by the individual light colors within the underlying PCMs (see *Figure 1a–b*), there is no direct proportionality. For instance, *Ac*PCM roughly assumes a 0.7:0.3 photostationary Pr:Pfr equilibrium under red light (see *Figure 1a*), but at the level of gene expression, the response of the *Ac*NIRusk circuit to red light is only 2% that of the NIR-light response. Initially unexpected, this baffling effect therefore further benefits the application of the pNIRusk circuits in combination with other light-responsive entities, be it optogenetic circuits or fluorescent reporters (*Meier et al., 2024b*; *Tabor et al., 2011*).

Our work provides pointers to the potential molecular underpinnings in that we isolated additional pNIRusk variants with similar NIR-light responses but substantially higher expression under red light (see *Figure 2b*). Notably, these variants only differ in the composition of their linkers connecting PCM to effector. Informed by our recent investigation of linker variants that culminated in the generation of *Dm*DERusk (*Meier et al., 2024a*), we hypothesize that the different red-light responses find their origin in an altered balance between the counteracting elementary histidine kinase and phosphatase activities of SHKs. At the ensemble level, even small perturbations, for example, caused by linker variation or changes in the Pr:Pfr proportion, can utterly tilt the net activity in either the kinase or phosphatase direction (*Landry et al., 2018*; *Möglich, 2019*; *Möglich et al., 2009*; *Russo and Silhavy, 1993*). The histidine kinase-phosphatase duality of TCSs thus provides a means of rendering the signal response to light highly malleable and adaptable. At a practical level, when engineering light-responsive TCSs, we advise to screen multiple candidate variants that may well differ substantially in their signal response. As noted previously (*Meier et al., 2024a*), similar concepts may also operate for certain cyclase (GGDEF) and phosphodiesterase (EAL) enzymes that frequently occur in conjunction with BphP PCMs (*Gourinchas et al., 2017*) and that antagonistically catalyze the making and breaking, respectively, of the bacterial second messenger cyclic-di-guanosine monophosphate (*Jenal et al., 2017*).

## Applications in the NIR future

The advent of stringently controlled NIR-responsive bacterial gene-expression platforms now facilitates a number of innovative applications. First, the currently available optogenetic setups for expression regulation in bacteria predominantly respond to blue and red light but are often blind to NIR light (*Ohlendorf and Möglich, 2022*). Given their fully intended relative insensitivity and their unexpectedly weak response to visible light, the pNIRusk systems lend themselves to combinations with pertinent setups. Applied at low to moderate intensity, blue light would trigger given setups, for example, pAurora2 or EL222-based systems (*Jayaraman et al., 2016*; *Ranzani et al., 2024*), but leave pNIRusk largely untouched. By contrast, NIR light would selectively activate pNIRusk as the LOV receptors underlying the blue-light-responsive setups are insensitive to this light quality. To facilitate multiplexed applications of pNIRusk, we supply derivatives reliant on alternative TCSs that can be readily combined in single bacterial cells and mediate the expression of several target genes orchestrated by two light colors.

Second, NIR light penetrates mammalian tissue much more deeply than light of shorter wavelengths (*Lehtinen et al., 2021*; *Weissleder, 2001*), owing to reduced absorption and scattering which are, for example, typically 5–10 lower for NIR light than for blue light. The exponential attenuation of light intensity with tissue depth is therefore much less pronounced for NIR light than it is for UV or visible light. Given the low phototoxicity of NIR light and its established use in photobiomodulation, noted above, the pNIRusk setups stand to be activatable through substantial tissue layers of at least several millimeters. Although the pNIRusk setups are (deliberately) less light-sensitive than, for example, the earlier pREDusk and pDERusk systems, they can be nonetheless actuated by light fluences in the range of several tens of µW per cm$^2$, which is orders of magnitudes less than the light intensities commonly used in neuronal optogenetics. These aspects open up intriguing applications involving programmable, light-responsive bacteria as agents inside the bodies of mammals, for example, as living therapeutics (*Cubillos-Ruiz et al., 2021*; *Hartsough et al., 2020*). The pDawn system has already been applied in this capacity within probiotic bacteria, but the approach has been hampered by delivering blue light in situ (*Cui et al., 2021*; *Magaraci et al., 2014*; *Pan et al., 2021*; *Sankaran and del Campo, 2019*; *Yang et al., 2020*). One remedy has been the use of upconverting nanoparticles (UCNP) administered alongside the light-responsive bacteria (*Pan et al., 2021*; *Yang et al., 2020*). Notably, the UCNPs can be charged by absorption of several photons in the NIR range to then emit a photon in the blue spectral region. Though feasible, pertinent applications introduce complexity and depart from a purely genetically encoded optogenetic strategy. The new pNIRusk setups directly address this challenge as they can be activated by wavelengths closely similar to those previously used for charging the UCNPs but at around 10-fold lower light intensity. Towards this goal, we demonstrate at present that both *Ac*NIRusk and *Av*NIRusk operate in the probiotic *E. coli* Nissle strain out of the box, without need for optimization. Used in place of pDawn, the pNIRusk systems thus obviate the need for the UCNPs and thereby broaden the application scope of living therapeutics considerably.

Third, we demonstrate the functionality of pNIRusk and pREDusk in the technologically relevant *A. tumefaciens* (*Figure 4c–d*) which is widely used as a transfection vehicle in plant biotechnology (*Liu et al., 2013*). Prospectively, both pNIRusk and pREDusk may achieve spatiotemporal control of *A. tumefaciens* infectivity, for example, to confine plant transfection to specific time windows or organs. Of particular advantage, the bathochromically shifted absorbance spectra of BphPs compared to plant Phys (*Hughes and Winkler, 2024*; *Yi et al., 2025*) stand to enable NIR-light control without interfering with plant photoreception and photoautotrophic growth.

# Materials and methods
## Molecular biology

Oligonucleotides and plasmids used in this study are listed in the *Supplementary file 1*. All plasmid constructs were confirmed by Sanger sequencing (Microsynth Seqlab, Göttingen, Germany). To produce bathy-BphP PCMs, the respective genes were introduced into a pCDF-Duet plasmid (Novagen) already containing heme oxygenase 1 (HO1) from *Synechocystis* sp. for biliverdin supply (*Mukougawa et al., 2006*; *Stüven et al., 2019*). To this end, the genes encoding the PCMs from *A. caulinodans* (Uniprot A8HU76, residues 1–501), *A. vitis* (Uniprot B9JR96, residues 1–499), and *P. aeruginosa* (Uniprot Q9HWR3, residues 1–497) were amplified by PCR from the template plasmids pQX091, pQX086, and pQX037, respectively (*Xu et al., 2024*), followed by Gibson assembly (*Gibson et al., 2009*).

The initial *Ac*NIRusk and *Av*NIRusk constructs (denoted '0') were created by exchanging the *Dm*PCM in *Dm*REDusk (*Meier et al., 2024a*) for *Ac*PCM and *Av*PCM, respectively (*Figure 2—figure supplement 1*), via PCR amplification and Gibson assembly. The second amino acid in *Ac*PCM and *Av*PCM was changed from proline to threonine to maintain the polycistronic structure of the operon comprising HO1, the BphP-SHK, and the RR FixJ. To apply the PATCHY strategy (*Ohlendorf et al., 2016*), start constructs were generated by elongating the PCM-DHp/CA linkers in the initial *Ac*NIRusk and *Av*NIRusk plasmids as reported before (*Meier et al., 2024a*). The resulting constructs thus comprise the N-terminal portions of the parental BphPs from A. caulinodans (Uniprot A8HU76) and A. vitis (Uniprot B9JR96) up to and including residues 520 and 518, respectively, connected to the C-terminal fragment of *B. japonicum* FixL starting from residue 257 (Uniprot P23222). At the junction, a restriction site for *Ksp*AI and a one-nucleotide frameshift were inserted (*Figure 2—figure supplement 2*). Cloning was performed by two consecutive overhang-PCRs and blunt-end ligation.

The TtrSR TCS from *Shewanella baltica* was introduced into *Dm*REDusk (*Meier et al., 2024a*) by first replacing the FixK2 promoter with the 85-bp minimal promoter sequence PttrB185-269 (*Daeffler et al., 2017*) via PCR amplification and restriction cloning using *Bgl*II and *Xba*I sites. Next, the *B. japonicum* FixLJ TCS was exchanged for the *S. baltica* TtrSR (Genbank CP000891.1, loci Sbal195_3858 and Sbal195_3859) via Gibson assembly using as a PCR template a synthetic gene (Gene Art) with codons optimized for expression in *E. coli*. For conducting PATCHY, a start construct was made that includes residues 1–539 of the *Dm*BphP (Uniprot E8U3T3) and the *S. baltica* TtrSR TCS (Uniprot A9L3S5) starting at residue 407. Cloning was performed as described above except that an *Eco*32I restriction site and a two-nucleotide frameshift were placed at the junction between the two gene fragments (*Figure 3—figure supplement 1a*).

The cognate TodX promoter of the TodST TCS from *P. putida* was amplified by PCR from pMIR77 (*Ramos-González et al., 2002*) and inserted into pDusk (*Ohlendorf et al., 2012*) by restriction cloning with *Bgl*II and *Xba*I. Next, the TodST TCS was amplified from pMIR66 (*Ramos-González et al., 2002*) and fused to the *Bacillus subtilis* YtvA LOV domain within pDusk via Gibson cloning. The *Av*TodNIRusk PATCHY start construct was obtained by replacing YtvA-LOV with HO1 and the *Av*PCM (residues 1–518) via Gibson assembly. The resultant construct comprised TodS starting from residue 726 (Uniprot E0X9C7) and TodT. At the junction, a *Ksp*AI restriction site and a one-nucleotide frameshift were inserted (*Figure 3—figure supplement 1b*).

For multiplexing experiments, the *Ds*Red Express2 reporter gene (*Strack et al., 2008*) in *Dm*DERusk and *Av*NIRusk was replaced by the YPet reporter gene (*Nguyen and Daugherty, 2005*). Additionally, the kanamycin (Kan) resistance marker and the ColE1 origin of replication (ori) were exchanged for a streptomycin (Str) marker and a CloDF13 ori via Gibson assembly. A version of *Av*NIRusk containing a

multiple-cloning site rather than a fluorescent reporter was cloned by Gibson assembly and deposited at Addgene for distribution (accession number 235084).

To apply *Dm*REDusk and *Av*NIRusk in *A. tumefaciens*, the *Photorhabdus luminescence luxABCDE* operon from pBAM2-luxAE (*Dziuba et al., 2021*) was cloned into the miniTn5 delivery plasmid pBAM2 (Uebe, unpublished). The sequence was amplified via PCR, digested with *Nde*I/*Sac*I, and ligated into a similarly digested and Fast-AP-dephosphorylated pBAM2 vector to generate pBAM2-luxAE. Subsequently, the light-regulated TCS cassettes of *Dm*REDusk (*DrhemO-Dmpcm-BjfixL*) and *Av*NIRusk (*DrhemO-Avpcm-BjfixL*) were amplified. These genes were set under the control of the strong constitutive *M. gryphiswaldense* PmamDC promoter. The *B. japonicum* FixK2 promoter was also amplified by PCR. The amplified fragments were fused by overlap extension PCR and digested with *Nde*I/*Not*I as was the pBAM2-luxAE fragment. Ligation resulted in the reporter plasmids *Dm*REDlux and *Av*NIRlux in which the *luxABCDE* operon is under the control of the FixK2 promoter.

## Protein purification

For production of *Ac*PCM, *Av*PCM, and *Pa*PCM, *E. coli* LOBSTR cells (*Andersen et al., 2013*) were transformed with the respective pCDF-Duet-PCM/HO1 constructs. Gene expression and protein purification were performed as before (*Meier et al., 2024a*). Briefly, two flasks with 800 mL lysogeny broth (LB) medium each supplemented with 100 μg mL$^{-1}$ Str were inoculated with a 5-mL overnight culture, and bacteria were grown in baffled Erlenmeyer flasks at 37°C and 225 rpm agitation. At an optical density at 600 nm (OD$_{600}$) of around 0.7, the expression was induced by 1 mM isopropyl-β-thiogalactopyranoside (IPTG), and 0.5 mM δ-aminolevulinic acid was added to support biliverdin incorporation. The expression was performed at 16°C and 225 rpm agitation in darkness for around 65 h. All following steps were performed in green safe light. After harvesting the cells by centrifugation, they were resuspended in lysis buffer [50 mM tris(hydroxy-methyl)-amino-methane (Tris)/HCl, 20 mM NaCl, 10 mM imidazole, 1 mM β-mercaptoethanol, pH 8.0, protease inhibitor completeOne (Roche), 1 mg mL$^{-1}$ lysozyme]. Cells were lysed by sonification, and the supernatant of the centrifuged lysate was applied to a HiTrap TALON crude column loaded with cobalt-nitrilotriacetic resin (Cytiva). The column was washed with buffer (50 mM Tris/HCl, 20 mM NaCl, 1 mM β-mercaptoethanol, pH 8.0), and the His-tagged proteins were eluted by an imidazole gradient from 0 to 0.5 M. Fractions were analyzed by denaturing SDS polyacrylamide gel electrophoresis (PAGE) in the presence of 1 mM Zn$^{2+}$ to visualize covalent biliverdin incorporation (*Berkelman and Lagarias, 1986*) and pooled based on protein content. During an overnight dialysis at 4°C into buffer (50 mM Tris/HCl, 20 mM NaCl, 1 mM β-mercaptoethanol, pH 8.0), the His-tag was cleaved off by His$_6$-tagged TEV protease. The sample was applied twice to a gravity-flow IMAC column with HisPur cobalt resin (Thermo Fisher) using the same buffers as before. Cleavage of the His-tag and protein purity were analyzed by Zn-SDS-PAGE. Fractions were pooled, dialyzed overnight at 4°C against storage buffer (20 mM Tris/HCl, 20 mM NaCl, 10% [w/v] glycerol, 1 mM dithiothreitol, pH 8.0), concentrated via spin filtration, and stored at –80°C.

## UV/vis absorption spectroscopy

Absorption spectroscopy was carried out on a Cary60 UV/vis spectrophotometer (Agilent Technologies) at 25°C under green safe light. Protein samples were diluted into buffers with pH ranging from 6.0 to 11.0 (30 mM 2-(N-morpholino)ethanesulfonic acid/HCl, 30 mM NaCl, pH 6.0/6.5/7.0; 30 mM Tris/HCl, 30 mM NaCl, pH 7.5/8.0/8.5/9.0; 30 mM glycine/HCl, 30 mM NaCl, pH 10.0/10.5/11.0; *Rumfeldt et al., 2019*). Samples were illuminated for 120 s with LEDs emitting in the NIR [(800±18) nm], red [(650±10) nm] and blue [(470±5) nm]. Data were normalized by the Soret absorbance at 403 nm. Pr→Pfr dark recovery kinetics were monitored every 5 s at 760 nm and fitted to single-exponential functions using the Fit-o-mat software (*Möglich, 2018*). Pr:Pfr ratios after red or blue illumination were calculated via linear combination of the basis spectra as described previously (*Yi et al., 2025*). The variation of photochemical parameters with pH was evaluated by fitting to the Henderson–Hasselbalch equation. The ratio of the quantum yields $\Phi_p$ and $\Phi_q$ for the red-light-driven photoconversion in the Pr→Pfr and Pfr→Pr directions, respectively (see *Figure 1—figure supplement 1f*), can be determined as follows (*Yi et al., 2025*):

$$\Phi_q/\Phi_p = P_R/P_{FR} \times \left(1 - 10^{-A_{Pr}}\right) / \left(1 - 10^{-A_{Pfr}}\right)$$

$P_R$ and $P_{FR}$ are the Pr and Pfr proportions at red-light-induced photostationary state, and $A_{Pr}$ and $A_{Pfr}$ denote the absorbance in the Pr and Pfr states at the peak emission wavelength of the red-light source.

## Generation and screening of linker variants

*Ac*NIRusk, *Av*NIRusk, *Dm*Ttr-REDusk, and *Av*Tod-NIRusk linker variants were created with the PATCHY strategy (**Meier et al., 2024a**; **Ohlendorf et al., 2016**). Sets of forward and reverse primers to generate linker variants were designed using a custom Python script (https://github.com/vrylr/PATCHY, copy archived at **vrylr, 2015**). In brief, PCR amplification was carried out using the primer sets in a final concentration of 0.25 µM. After purifying the PCR product by gel extraction, the start constructs were digested with *Ksp*AI or *Eco*32I (**Figure 2—figure supplement 2**, **Figure 3—figure supplement 1**), followed by phosphorylation with T4 polynucleotide kinase and ligation with T4 DNA ligase. The reaction mixes were transformed into DH10b cells and plated on LB plates supplemented with kanamycin (LB/Kan). Plates were incubated overnight at 37°C in darkness, or under red light [100 µW cm$^{-2}$, (650±11) nm] or NIR light [200 µW cm$^{-2}$, (800±18) nm]. Applied light intensities were calibrated with a power meter (model 842-PE, equipped with a 918D-UV-OD3 silicon photodiode, Newport, Darmstadt, Germany). Bacterial colonies developing red color upon overnight incubation, indicative of *Ds*Red production, were used to inoculate 200 µL LB/Kan in a clear 96-well microtiter plate (MTP) (Nunc, Thermo Fisher). Bacteria were incubated for 21–24 h at 30°C and 750 rpm agitation. Next, the cultures were diluted 100-fold into fresh LB/Kan medium and split into three separate cultures which were then incubated at 37°C and 750 rpm agitation in either darkness, red light [100 µW cm$^{-2}$, (650±11) nm], or NIR light [200 µW cm$^{-2}$, (800±18) nm]. After 18 h, the optical density at (600±9) nm ($OD_{600}$) and the *Ds*Red fluorescence [(554±9) nm excitation and (591±20) nm emission] were measured with a Tecan Infinite M200 Pro MTP reader (Tecan Group, Ltd., Männedorf, Switzerland). The *Ds*Red fluorescence was normalized by $OD_{600}$ and represent the mean ± s.d. of three biologically independent replicates.

## Light-dose–response assays in *Escherichia coli*

Analyses were performed as before (**Meier et al., 2024a**; **Multamäki et al., 2022**). *E. coli* CmpX13 (**Mathes et al., 2009**) or *E. coli* Nissle (*Ec*N, DSMZ no. 115365) bacteria containing the respective construct were used to inoculate 5 mL LB medium supplemented with 50 µg mL$^{-1}$ Kan (LB/Kan) or 100 µg mL$^{-1}$ Str (LB/Str). Cultures were incubated for 21–24 h at 30°C and 225 rpm agitation under non-inducing light conditions. Under green safe light, the bacteria were diluted 100-fold into fresh LB/Kan or LB/Str, before distributing 200 µL each into 64 wells of a black-walled MTP with clear bottom (µClear, Greiner BioOne, Frickenhausen, Germany). As a control, bacteria harboring an empty plasmid variant with a multiple-cloning site (MCS) were also placed on the MTP. After sealing the MTP with a gas-permeable membrane, it was placed on an Arduino-controlled illumination device. For the application of red and blue light, a previous setup was used (**Hennemann et al., 2018**) with LED peak wavelengths of (624±8) nm and (463±12) nm, respectively. To apply NIR light, we devised a new matrix of 8-by-8 LEDs emitting at (800±18) nm (LED800-01AU, Roithner Lasertechnik) which is controlled by an Arduino circuit board (**Figure 2—figure supplement 3**). The bacteria inside the MTPs were incubated for 18 h at 37°C and 750 rpm agitation. The illumination was applied in 1:2 or 1:10 duty cycles (i.e., 1 s light and 1 s darkness, or 20 s light and 180 s darkness). After incubation, the $OD_{600}$ and *Ds*Red fluorescence were determined as above. For measurements of YPet fluorescence, excitation and emission wavelengths of (500±9) nm and (554±9) nm were used. Following normalization to $OD_{600}$, the MCS background was subtracted. Data represent mean ± s.d. of three biological replicates and were evaluated as a function of the light intensity averaged over the duty cycle according to Hill isotherms using Fit-o-mat.

## Flow cytometry

*E. coli* CmpX13 cells (**Mathes et al., 2009**) containing *Ac*NIRusk, *Av*NIRusk, or an empty-vector control (MCS) were analyzed by flow cytometry (**Meier et al., 2024a**). After overnight incubation for 21–24 h at 30°C and 750 rpm agitation, bacterial cultures were diluted 100-fold into fresh LB/Kan medium. For each variant, the cultures were split and incubated for 18 h at 37°C and 750 rpm agitation in either darkness, red light [100 µW cm$^{-2}$, (624±8) nm], blue light [70 µW cm$^{-2}$, (463 ± 12) nm], or

different intensities (18–270 µW cm$^{-2}$) of NIR light [(800 ± 18) nm]. Following incubation, the cultures were diluted 10-fold into phosphate-buffered saline (1x sheath fluid, Bio-Rad) and analyzed on a S3e cell sorter (Bio-Rad). The $D$sRed fluorescence of at least 200,000 events was detected using excitation lasers at 488 nm and 561 nm and an emission channel at (585 ± 15) nm. Data were fitted to skewed Gaussian distributions using Fit-o-mat. Experiments were repeated three times with similar results.

## Growth and expression kinetics

The growth and reporter-gene expression of *E. coli* CmpX13 cells (*Mathes et al., 2009*) containing *Ac*NIRusk, *Av*NIRusk, or an empty-vector control (MCS) were analyzed over time as described before (*Multamäki et al., 2022*). For each variant, two 5-mL LB/Kan cultures were inoculated and incubated in darkness and under NIR light [200 µW cm$^{-2}$, (800±18) nm] at 37°C and 600 rpm agitation. At discrete times, samples of 200 µL were supplemented with 3.5 mg mL$^{-1}$ chloramphenicol and 0.4 mg mL$^{-1}$ tetracycline to arrest cell growth and translation. After an incubation of at least 2 h, the $OD_{600}$ and $D$sRed fluorescence were determined as above. Data represent mean ± s.d. of three biological replicates and were fitted to logistic functions.

## Multiplexing of optogenetic circuits

Pairs of plasmids harboring light-responsive TCSs were serially transformed into chemically competent *E. coli* CmpX13. 200 µL LB supplemented with 50 µg mL$^{-1}$ Kan and 100 µg mL$^{-1}$ Str (LB/Kan+Str) were inoculated with single bacterial clones and incubated for 21–24 h inside MTP wells at 30°C and 750 rpm agitation in darkness. The cultures were then diluted 100-fold into LB/Kan+Str before dispensation of 200 µL each into individual wells of four µClear MTPs. The MTPs were incubated for 18 h at 37°C and 750 rpm agitation in either darkness, under red light [100 µW cm$^{-2}$, (624±8) nm], NIR light [200 µW cm$^{-2}$, (800±18) nm], or both light colors combined. Where applicable, red light was applied continuously from below using the Arduino-controlled RGB LED matrix, whereas NIR light shone from above. After 18 h incubation, the $OD_{600}$, and $D$sRed and YPet fluorescence were measured as above. Data represent mean ± s.d. of three biological replicates.

## Luciferase assays in *A. tumefaciens*

The *Dm*REDlux and *Av*NIRlux plasmids were transferred to *A. tumefaciens* C58 (also known as *Agrobacterium fabrum* C58, Lab collection, ATCC 33970) by biparental conjugation using *E. coli* WM3064 as a diaminopimelic acid auxotrophic donor strain (William Metcalf, UIUC, unpublished; *thrB1004 pro thi rpsL hsdS lacZΔM15 RP4-1360 Δ(araBAD)567 ΔdapA1341::[erm pir (wt)]*). Unless specified otherwise, C58 was routinely cultivated in LB medium at 28°C with 170 rpm agitation. Selection for mini-Tn5 insertion mutants carrying the *lux* reporter constructs was carried out on solid LB with 1.5% (w/v) agar and 50 µg mL$^{-1}$ Kan. *E. coli* WM3064 strains carrying plasmids were cultivated at 37°C with 200 rpm shaking in LB supplemented with 0.1 mM DL-a,e-diaminopimelic acid (DAP) and 25 µg mL$^{-1}$ Kan or 50 µg mL$^{-1}$ ampicillin (Amp).

The protocol for biparental conjugation was described before (*Dziuba et al., 2021*). Briefly, cultures of the C58 receptor and plasmid-containing *E. coli* strains were cultivated in LB or LB/Amp medium overnight. Then, 2×10$^9$ cells of each strain were mixed in 15 mL falcon tubes and pelleted at 5421 × *g* for 10 min. The supernatant was discarded while the cell pellet was resuspended in 100–200 µL LB medium and spotted onto LB agar plates. After overnight incubation at 28°C, cells were resuspended in 1 mL LB and transferred to 15 mL falcon tubes containing 9 mL LB medium. After incubation for 2 h at 28°C and 120 rpm agitation, the cell suspension was pelleted by centrifugation, and the supernatant was discarded. Following resuspension in 1 mL LB medium, 100 µL of 10- and 100-fold diluted cell suspensions were plated on LB-Kan agar plates. After incubation for three days at 28°C, Kan-resistance plasmid insertion mutants were transferred to fresh LB/Kan agar plates and screened by PCR.

At least three randomly selected transconjugants containing the luciferase reporter constructs were analyzed for luminescence. Each experiment was performed in duplicate. Luminescence signals, measured as arbitrary light units (ALU), were detected using a Tecan Infinite M200 Pro MTP reader equipped with a luminometer module (Tecan Group, Ltd., Männedorf, Switzerland) during culture growth in LB medium at 23°C and 280 rpm. Measurements were taken every 20 min over 300 cycles (~90 h). ALU readings were normalized by the optical density at 650 nm ($OD_{650}$) to calculate the

relative light units (RLU). To this end, tested clones were inoculated into 600 µL LB/Kan medium in 24-well plates and incubated for ~20 h under non-inducing conditions (i.e., red light for *Dm*REDlux and darkness for *Av*NIRlux, respectively) at 28°C and 140 rpm. Cultures were then diluted 10-fold into 1 mL LB/Kan and incubated under identical conditions for an additional 4–5 h. Subsequently, the $OD_{650}$ was adjusted to 0.1 by adding fresh LB/Kan medium, and 100 µL of the normalized cell suspension was transferred into two 96-well MTPs. Cultivation of both MTPs was initiated simultaneously, with one incubated under dark conditions and the other one under red [(650±11) nm] or NIR light [(800±18) nm].

Alternatively, luminescence was assessed using a spotting assay. Precultures were prepared as above, and 5 µL of the $OD_{650}$-normalized cell suspension were spotted onto LB/Kan agar plates. The plates were incubated for 44 h at 28°C under dark conditions or under blue [(470±13) nm], red [(650±11) nm], or NIR light [(800±18) nm]. Luminescence signals were documented as cumulative signal intensities using the Bio-Rad ChemiDoc XRS+ System (Bio Rad Laboratories Inc, Herkules, USA), with images captured every 0.6 s over a 1 min period. In addition, colorimetric images were acquired to visualize the outlines of the spotted cell suspensions. Luminescence was quantified from the image time series by calculating the slope of the mean pixel intensities within the outlined cell suspension areas using the ImageJ Fiji package (v1.54f) (*Schindelin et al., 2012*). Background correction was performed by subtracting the mean pixel intensity of an uninoculated region of the agar plate.

## Acknowledgements

We thank our colleagues for discussion; Dr. Roman S Iwasaki, Dr. Ralph P Diensthuber, and Tobias von Basse for preliminary experiments; Tino Krell for generously providing the pMIR66 and pMIR77 plasmids; and Mr. Braun and the electronic workshop at the University Bayreuth for support. Funding was provided by the European Commission (FET Open NEUROPA grant 863214 to AM); the German Research Foundation (grants MO2192/4-2 to AM and UE200/1-1 to RU); and the Research Council of Finland (grant 330678 to HT).

## Additional information

### Funding

| Funder | Grant reference number | Author |
|---|---|---|
| European Commission | 10.3030/863214 | Andreas Möglich |
| Deutsche Forschungsgemeinschaft | MO2192/4-2 | Andreas Möglich |
| Deutsche Forschungsgemeinschaft | UE200/1-1 | René Uebe |
| Research Council of Finland | 330678 | Heikki Takala |

The funders had no role in study design, data collection and interpretation, or the decision to submit the work for publication.

### Author contributions

Stefanie SM Meier, Conceptualization, Data curation, Formal analysis, Investigation, Visualization, Methodology, Writing – original draft, Writing – review and editing; Michael Hörzing, Cornelia Böhm, Emma LR Düthorn, Data curation, Formal analysis, Investigation, Writing – review and editing; Heikki Takala, Funding acquisition, Writing – review and editing; René Uebe, Data curation, Formal analysis, Funding acquisition, Investigation, Writing – review and editing; Andreas Möglich, Conceptualization, Data curation, Formal analysis, Supervision, Funding acquisition, Investigation, Writing – original draft, Project administration, Writing – review and editing

### Author ORCIDs

Stefanie SM Meier https://orcid.org/0009-0002-6028-5223
Cornelia Böhm https://orcid.org/0000-0002-3552-2349

Heikki Takala https://orcid.org/0000-0003-2518-8583
René Uebe https://orcid.org/0000-0003-2357-1589
Andreas Möglich https://orcid.org/0000-0002-7382-2772

Reviewer #1 (Public review): https://doi.org/10.7554/eLife.107069.3.sa1
Reviewer #2 (Public review): https://doi.org/10.7554/eLife.107069.3.sa2
Reviewer #3 (Public review): https://doi.org/10.7554/eLife.107069.3.sa3
Author response https://doi.org/10.7554/eLife.107069.3.sa4

## Additional files

### Supplementary files
MDAR checklist

Supplementary file 1. Information on oligonucleotide primers (**A**) and plasmids (**B**) used in this study.

### Data availability
Data underlying the main figures and figure supplements of the work are available at Zenodo (https://doi.org/10.5281/zenodo.16933398). The AvNIRusk plasmid is available from Addgene under accession number 235084. Additional materials are available from the corresponding author upon reasonable request.

The following dataset was generated:

| Author(s) | Year | Dataset title | Dataset URL | Database and Identifier |
| --- | --- | --- | --- | --- |
| Meier S, Möglich A | 2025 | Experimental data from Meier et al. (2025) Engineering NIR-Sighted Bacteria | https://doi.org/10.5281/zenodo.16933398 | Zenodo, 10.5281/zenodo.16933398 |

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
