## [Editor Report · eLife Assessment]

This **important** study establishes bathy phytochromes, a unique class of bacterial photoreceptors that respond to near-infrared light (NIR), as versatile tools for bacterial optogenetics. NIR light is a key control signal in optogenetics due to its deep tissue penetration and the ability to combine with existing red- and blue-light sensitive systems, but thus far, NIR-activated proteins have been poorly characterized. The strength of evidence is **convincing**, with comprehensive in vitro characterization, modular design strategies, and validation across different hosts, supporting the versatility and potential for these tools in biotechnological applications. This study should advance the fields of optogenetics and photobiology and inspire future work.

---

## [Referee Report · Reviewer #1 (Public review)]

Summary:

This is an interesting study characterizing and engineering so-called bathy phytochromes, i.e. those that respond to near infrared (NIR) light in the ground state, for optogenetic control of bacterial gene expression. Previously, the authors have developed a structure-guided approach to functionally link several light responsive protein domains to the signaling domain of the histidine kinase FixL, which ultimately controls gene expression. Here, the authors use the same strategy to link bathy phytochrome light responsive domains to FixL, resulting in sensors of NIR light. Interestingly, they also link these bathy phytochrome light sensing domains to signaling domains from the tetrathionate-sensing SHK TtrS and the toluene-sensing SHK TodS, demonstrating generality of their protein engineering approach more broadly across bacterial two-component systems.

This is an exciting result that should inspire future bacterial sensor design. The authors go on to leverage this result to develop what is, to my knowledge, the first system for orthogonally controlling the expression of two separate genes in the same cell with NIR and Red light, a valuable contribution to the field.

Finally, the authors reveal new details of the pH-dependent photocycle of bathy phytochromes and demonstrate their sensors work in the gut- and plant-relevant strains *E. coli* Nissle 1917 and A. tumefaciens.

Strengths:

The experiments are well founded, well executed, and rigorous.

The manuscript is clearly written.

The sensors developed exhibit large responses to light, making them valuable tools for ontogenetic applications.

This study is a valuable contribution to photobiology and optogenetics.

Weaknesses:

As the authors note, the sensors are relatively insensitive to NIR light due to the rapid dark reversion process in bathy phytochromes. Though NIR light is generally non-phototoxic, one would expect this characteristic to be a limitation in some downstream applications where light intensities are not high (e.g. in vivo).

Though they can be multiplexed with Red light sensors, these bathy phytochrome NIR sensors are more difficult to multiplex with other commonly used light sensors (e.g. blue) due to the broad light responsivity of the Pfr state. This challenge may be overcome by careful dosing of blue light, as the authors discuss, but other bacterial NIR sensing systems with less cross-talk may be preferred in some applications.

Comments on revisions:

My concerns have been addressed.

---

## [Referee Report · Reviewer #2 (Public review)]

In this manuscript, Meier et al. engineer a new class of light-regulated two-component systems. These systems are built using bathy-bacteriophytochromes that respond to near-infrared (NIR) light. Through a combination of genetic engineering and systematic linker optimization, the authors generate bacterial strains capable of selective and tunable gene expression in response to NIR stimulation. Overall, these results are an interesting expansion of the optogenetic toolkit into the NIR range. The cross-species functionality of the system, modularity, and orthogonality have the potential to make these tools useful for a range of applications.

Strengths:

(1) The authors introduce a novel class of near-infrared light-responsive two-component systems in bacteria, expanding the optogenetic toolbox into this spectral range.

(2) Through engineering and linker optimization, the authors achieve specific and tunable gene expression, with minimal cross-activation from red light in some cases.

(3) The authors show that the engineered systems function robustly in multiple bacterial strains, including laboratory *E. coli*, the probiotic *E. coli* Nissle 1917, and Agrobacterium tumefaciens.

(4) The combination of orthogonal two-component systems can allow for simultaneous and independent control of multiple gene expression pathways using different wavelengths of light.

(5) The authors explore the photophysical properties of the photosensors, investigating how environmental factors such as pH influence light sensitivity.

Comments on revisions:

The authors have addressed all my prior concerns.

---

## [Referee Report · Reviewer #3 (Public review)]

Summary:

This paper by Meier et al introduces a new optogenetic module for regulation of bacterial gene expression based on "bathy-BphP" proteins. Their paper begins with a careful characterization of kinetics and pH dependence of a few family members, followed by extensive engineering to produce infrared-regulated transcriptional systems based on the authors' previous design of the pDusk and pDERusk systems, and closing with characterization of the systems in bacterial species relevant for biotechnology.

Strengths:

The paper is important from the perspective of fundamental protein characterization, since bathy-BphPs are relatively poorly characterized compared to their phytochrome and cyanobacteriochrome cousins. It is also important from a technology development perspective: the optogenetic toolbox currently lacks infrared-stimulated transcriptional systems. Infrared light offers two major advantages: it can be multiplexed with additional tools, and it can penetrate into deep tissues with ease relative to the more widely used blue light activated systems. The experiments are performed carefully and the manuscript is well written.

Weaknesses:

Some of the light-inducible responses described in this compelling paper are complex and difficult to rationalize, such as the dependence of light responses on linker length and differences in responses observed from the bathy-BphPs in isolation versus strains in which they are multiplexed. Nevertheless, the authors should be commended for carrying out rigorous experiments and reporting these results accurately. These are minor weaknesses in an overall very strong paper.

---

## [Author Response]

The following is the authors’ response to the original reviews.

**Reviewer #1 (Public review):**
Summary:This is an interesting study characterizing and engineering so-called bathy phytochromes, i.e., those that respond to near infrared (NIR) light in the ground state, for optogenetic control of bacterial gene expression. Previously, the authors have developed a structure-guided approach to functionally link several light-responsive protein domains to the signaling domain of the histidine kinase FixL, which ultimately controls gene expression. Here, the authors use the same strategy to link bathy phytochrome light-responsive domains to FixL, resulting in sensors of NIR light. Interestingly, they also link these bathy phytochrome light-sensing domains to signaling domains from the tetrathionate-sensing SHK TtrS and the toluene-sensing SHK TodS, demonstrating the generality of their protein engineering approach more broadly across bacterial two-component systems.This is an exciting result that should inspire future bacterial sensor design. They go on to leverage this result to develop what is, to my knowledge, the first system for orthogonally controlling the expression of two separate genes in the same cell with NIR and Red light, a valuable contribution to the field.Finally, the authors reveal new details of the pH-dependent photocycle of bathy phytochromes and demonstrate that their sensors work in the gut - and plant-relevant strains *E. coli* Nissle 1917 and A. tumefaciens.Strengths:(1) The experiments are well-founded, well-executed, and rigorous.(2) The manuscript is clearly written.(3) The sensors developed exhibit large responses to light, making them valuable tools for ontogenetic applications.(4) This study is a valuable contribution to photobiology and optogenetics.

We thank the reviewer for the positive verdict on our manuscript.

Weaknesses:(1) As the authors note, the sensors are relatively insensitive to NIR light due to the rapid dark reversion process in bathy phytochromes. Though NIR light is generally non-phototoxic, one would expect this characteristic to be a limitation in some downstream applications where light intensities are not high (e.g., in vivo).

We principally concur with this reviewer’s assessment that delivery of light (of any color) into living tissue can be severely limited by absorption, reflection, and scattering. That notwithstanding, at least two considerations suggest that in-vivo deployment of the pNIRusk setups we presently advance may be feasible.

First, while the pNIRusk setups are indeed less light-sensitive compared to, e.g., our earlier redlight-responsive pREDusk and pDERusk setups (see Meier et al. Nat Commun 2024), we note that the overall light fluences required for triggering them are in the range of tens of µW per cm_2_. By contrast, optogenetic experiments in vivo, in particular in the neurosciences, often employ light area intensities on the order of mW per cm_2_ and above. Put another way, compared to the optogenetic tools used in these experiments, the pNIRusk setups are actually quite sensitive to light.

Second, sensitivity to NIR light brings the advantage of superior tissue penetration, see data reported by Weissleder Nat Biotech 2001 and Ash et al. Lasers Med Sci 2017 (both papers are cited in our manuscript). Based on these data, the intensity of blue light (450 nm) therefore falls off 5-10 times more strongly with penetration depth than that of NIR light (800 nm).

We have added a brief treatment of these aspects in the Discussion section.

(2) Though they can be multiplexed with Red light sensors, these bathy phytochrome NIR sensors are more difficult to multiplex with other commonly used light sensors (e.g., blue) due to the broad light responsivity of the Pfr state. This challenge may be overcome by careful dosing of blue light, as the authors discuss, but other bacterial NIR sensing systems with less cross-talk may be preferred in some applications.

The reviewer is correct in noting that, at least to a certain extent, the pNIRusk systems also respond to blue light owing to their Soret absorbance bands (see Fig. 1). That said, we note two points:

First, a given photoreceptor that preferentially responds to certain wavelengths, e.g., 700 nm in the case of conventional bacterial phytochromes (BphP), generally absorbs shorter wavelengths to some degree as well. Absorption of these shorter wavelengths suffices for driving electronic and/or vibronic transitions of the chromophore to higher energy levels which often give rise to productive photochemistry and downstream signal transduction. Put another way, a certain response of sensory photoreceptors to shorter wavelengths is hence fully expected and indeed experimentally borne out, as for instance shown by Ochoa-Fernandez et al. in the so-called PULSE setup (Nat Meth 2020, doi: 10.1038/s41592-020-0868-y).

Second, known BphPs share similar Pr and Pfr absorbance spectra. We therefore expect other BphP-based optogenetic setups to also respond to blue light to some degree. Currently, there are insufficient data to gauge whether individual BphPs systematically differ in their relative sensitivity to blue compared to red or NIR light. Arguably, pertinent experiments may be an interesting subject for future study.

**Reviewer #2 (Public review):**
Summary:In this manuscript, Meier et al. engineer a new class of light-regulated two-component systems. These systems are built using bathy-bacteriophytochromes that respond to near-infrared (NIR) light. Through a combination of genetic engineering and systematic linker optimization, the authors generate bacterial strains capable of selective and tunable gene expression in response to NIR stimulation. Overall, these results are an interesting expansion of the optogenetic toolkit into the NIR range. The cross-species functionality of the system, modularity, and orthogonality have the potential to make these tools useful for a range of applications.Strengths:(1) The authors introduce a novel class of near-infrared light-responsive two-component systems in bacteria, expanding the optogenetic toolbox into this spectral range.(2) Through engineering and linker optimization, the authors achieve specific and tunable gene expression, with minimal cross-activation from red light in some cases.(3) The authors show that the engineered systems function robustly in multiple bacterial strains, including laboratory *E. coli*, the probiotic *E. coli* Nissle 1917, and Agrobacterium tumefaciens.(4) The combination of orthogonal two-component systems can allow for simultaneous and independent control of multiple gene expression pathways using different wavelengths of light.(5) The authors explore the photophysical properties of the photosensors, investigating how environmental factors such as pH influence light sensitivity.Weaknesses:(1) The expression of multi-gene operons and fluorescent reporters could impose a metabolic burden. The authors should present data comparing optical density for growth curves of engineered strains versus the corresponding empty-vector control to provide insight into the burden and overall impact of the system on host viability and growth.

In response to this comment, we have recorded growth kinetics of bacteria harboring the pNIRusk-DsRed plasmids or empty vectors under both inducing (i.e., under NIR light) and noninducing conditions (i.e., darkness). We did not observe systematic differences in the growth kinetics between the different cultures, thus suggesting that under the conditions tested there is no adverse effect on cell viability.

We include the new data in Suppl. Fig. 5c-d and refer to them in the main text.

(2) The manuscript consistently presents normalized fluorescence values, but the method of normalization is not clear (Figure 2 caption describes normalizing to the maximal fluorescence, but the maximum fluorescence of what?). The authors should provide a more detailed explanation of how the raw fluorescence data were processed. In addition, or potentially in exchange for the current presentation, the authors should include the raw fluorescence values in supplementary materials to help readers assess the actual magnitude of the reported responses.

We appreciate this valid comment and have altered the representation of the fluorescence data. All values for a given fluorescent protein (i.e., either DsRed or YPet) across all systems are now normalized to a single reference value, thus enabling direct comparison between experiments.

(3) Related to the prior point, it would be useful to have a positive control for fluorescence that could be used to compare results across different figure panels.

As all data are now normalized to the same reference value, direct comparison across all figures is enabled.

(4) Real-time gene expression data are not presented in the current manuscript, but it would be helpful to include a time-course for some of the key designs to help readers assess the speed of response to NIR light.

In response to this comment, we include in the revised manuscript induction kinetics of bacterial cultures bearing pNIRusk upon transfer to inducing NIR-light conditions. To this end, aliquots were taken at discrete timepoints, transcriptionally and translationally arrested, and analyzed for optical density and DsRed reporter fluorescence after allowing for chromophore maturation.

We include the new data in Suppl. Fig. 5e and refer to them in the manuscript.

Moreover, we note that the experiments in Agrobacterium tumefaciens used a luciferase reporter thus enabling the continuous monitoring of the light-induced expression kinetics. These data (unchanged in revision) are to be found in Suppl. Fig. 9.

**Reviewer #3 (Public review):**
Summary:This paper by Meier et al introduces a new optogenetic module for the regulation of bacterial gene expression based on "bathy-BphP" proteins. Their paper begins with a careful characterization of kinetics and pH dependence of a few family members, followed by extensive engineering to produce infrared-regulated transcriptional systems based on the authors' previous design of the pDusk and pDERusk systems, and closing with characterization of the systems in bacterial species relevant for biotechnology.Strengths:The paper is important from the perspective of fundamental protein characterization, since bathyBphPs are relatively poorly characterized compared to their phytochrome and cyanobacteriochrome cousins. It is also important from a technology development perspective: the optogenetic toolbox currently lacks infrared-stimulated transcriptional systems. Infrared light offers two major advantages: it can be multiplexed with additional tools, and it can penetrate into deep tissues with ease relative to the more widely used blue light-activated systems. The experiments are performed carefully, and the manuscript is well written.Weaknesses:My major criticism is that some information is difficult to obtain, and some data is presented with limited interpretation, making it difficult to obtain intuition for why certain responses are observed. For example, the changes in red/infrared responses across different figures and cellular contexts are reported but not rationalized. Extensive experiments with variable linker sequences were performed, but the rationale for linker choices was not clearly explained. These are minor weaknesses in an overall very strong paper.

We are grateful for the positive take on our manuscript.

**Reviewer #1 (Recommendations for the authors):**
(1) As eLife is a broad audience journal, please define the Soret and Q-bands (line 125).

We concur and have added labels in fig. 1a that designate the Soret and Q bands.

(2) The initial (0) Ac design in Figure 2b is activated by NIR and Red light, albeit modestly. The authors state that this construct shows "constant reporter fluorescence, largely independent of illumination" (line 167). This language should be changed to reflect the fact that this Ac construct responds to both of these wavelengths.

Agreed. We have amended the text accordingly.

(3) pNIRusk Ac 0 appears to show a greater light response than pNIRusk Av -5. However, the authors claim that the former is not light-responsive and the latter is. This conclusion should be explained or changed.

The assignment of pNIRusk Av-5 as light-responsive is based on the relative difference in reporter fluorescence between darkness and illumination with either red or NIR light. Although the overall fluorescence is much lower in Av-5 than for Av-0, the relative change upon illumination is much more pronounced. We add a statement to this effect to the text.

(4) The authors state that "when combining DmDERusk-Str-YPet with AvTod+21-DsRed expression rose under red and NIR light, respectively, whereas the joint application of both light colors induced both reporter genes" (lines 258-261). In contrast, Figure 3c shows that application of both wavelengths of light results in exclusive activation of YPet expression. It appears the description of the data is wrong and must be corrected. That said, this error does not impact their conclusion that two separate target genes can be independently activated by NIR and red light.

We thank the reviewer for catching this error which we have corrected in the revised manuscript.

(5) Line 278: I don't agree with the authors' blanket statement that the use of upconversion nanoparticles is a "grave" limitation for NIR-light mediated activation of bacterial gene expression in vivo. The authors should either expound on the severity of the limitation or use more moderate language.

We have replaced the word ‘grave’ by ‘potential’ and thereby toned down our wording.

**Reviewer #2 (Recommendations for the authors):**
(1) Please include a discussion on the expected depth penetration of different light wavelengths. This is most relevant in the context of the discussion about how these NIR systems could be used with living therapeutics.

Given the heterogeneity of biological tissue, it is challenging to state precise penetration depths for different wavelengths of light. That said, blue light for instance is typically attenuated by biological tissue around 5 to 10 times as strongly as near-infrared light is.

We have expanded the Discussion chapter to cover these aspects.

(2) It would be helpful for Figure 2C (or supplementary) to also include the response to blue light stimulation.

We agree and have acquired pertinent data for the blue-light response. The new data are included in an updated Fig. 2c. Data acquired at varying NIR-light intensities, originally included in Fig. 2c, have been moved to Suppl. Fig. 5a-b.

(3) In Figure 4A, data on the response of *E. coli* Nissle to blue and red light are missing. Including this would help identify whether the reduced sensitivity to non-NIR wavelengths observed in the *E. coli* lab strain is preserved in the probiotic background.

In response to this comment, we have acquired pertinent data on *E. coli* Nissle. While the results were overall similar to those in the laboratory strain, the response to blue and NIR light was yet lower in the Nissle bacteria which stands to benefit optogenetic applications.

We have updated Fig. 4a accordingly. For clarity, we only show the data for AvNIRusk in the main paper but have relegated the data on AcNIRusk to Suppl. Fig. 8. (Note that this has necessitated a renumbering of the subsequent Suppl. Figs.)

(4) On many of the figures, there are thin gray lines that appear between the panels that it would be nice to eliminate because, in some cases, they cut through words and numbers.

The grey lines likely arose from embedding the figures into the text document. In the typeset manuscript, which has become available on the eLife webpage in the meantime, there are no such lines. That said, we will carefully check throughout the submission/publishing/proofing process lest these lines reappear.

(5) Page 7, line 155: "As not least seen" typo or awkward phrasing.

We have restructured the sentence and thereby hopefully clarified the unclear phrasing.

(6) Page 7, line 167: It does not appear to be the case that the initial pNIRusk designs show constant fluorescence that is largely independent of illumination. AcNIRusk shows an almost twofold change from dark to NIR. Reword this to avoid confusion.

We concur with this comment, similar to reviewer #1’s remark, and have adjusted the text accordingly.

(7) Page 8, line 174: Related to the previous point, AvNIRusk has one design that is very minimally light switchable (-5), so stating that six light switchable designs have been identified is also confusing.

As stated in our response to reviewer #1 above, the assignment of AvNIRusk-5 as light-switchable is based on the relative fluorescence change upon illumination. We have added an explanation to the text.

(8) Page 10, line 228-229: I was not able to find the data showing that expression levels were higher for the DmTtr systems than the pREDusk and pNIRusk setups. This may be an issue related to the normalization point. It was not clear to me how to compare these values.

We apologize for the initially unclear representation of the data. In response to this reviewer’s general comments above, we have now normalized all fluorescence values to a single reference value, thus allowing their direct comparison.

(9) Page 12, line 264: "finer-grained expression control can be exerted..." Either show data or adjust the language so that it is clear this is a prediction.

True, we have replaced ‘can’ by ‘could’.

(10) Page 25, line 590: CmpX13 cells have a reference that is given later, but it should be added where it first appears.

Agreed, we have added the reference in the indicated place.

(11) Page 25, line 592: define LB/Kan.

We had already defined this abbreviation further up but, for clarity, we have added it again in the indicated position.

(12) Page 40, line 946: "normalized by" rather than "to".

We have implemented the requested change in the indicated and several other positions of the manuscript.

(13) Figures 2C, 3C, and similar plots in the supplementary material would benefit from having a legend for the colors.

We agree and have added pertinent legends to the corresponding main and supplementary figures.

(14) As a reader, I had some trouble following all the acronyms. This is at the author's discretion, but I would eliminate ones that are not strictly essential (e.g. MTP for microtiter plate; I was unable to identify what "MCS" meant; look for other opportunities to remove acronyms).

In the revised manuscript, we have defined the abbreviation ‘MCS’ (for ‘multiple-cloning site’) upon first occurrence. We have decided to retain the abbreviation ‘MTP’ in the text.

(15) Could the authors briefly speculate on why A. tumefaciens activation with red light might occur?

While we can but speculate as to the underlying reasons for the divergent red-light response in A. tumefaciens, we discuss possible scenarios below.

Commonly, two-component systems (TCS) exhibit highly cooperative and steep responses to signal. As a consequence, even small differences in the intracellular amounts of phosphorylated and unphosphorylated response regulator (RR) can give to significantly changed gene-expression output. Put another way, the gene-expression output need not scale linearly with the extent of RR phosphorylation but, rather, is expected to show nonlinear dependence with pronounced thresholding effects.

Differences in the pertinent RR levels can for instance arise from variations in the expression levels of the pNIRusk system components between *E. coli* and *A. tumefaciens*. Moreover, the two bacteria greatly differ in their two-component-system (TCS) repertoire. Although TCSs are commonly well insulated from each other, cross-talk with endogenous TCSs, even if limited, may cause changes in the levels of phosphorylated RR and hence gene-expression output. In a similar vein, the RR can also be phosphorylated and dephosphorylated non-enzymatically, e.g., by reaction with high-energy anhydrides (such as acetyl phosphate) and hydrolysis, respectively. Other potential origins for the divergent red-light response include differences in the strength of the promoters driving expression of the pNIRusk system components and the fluorescent/luminescent reporters, respectively.

(16) It would be helpful for the authors to briefly explain why they needed to switch to luminescence from fluorescence for the A. tumeraciens studies.

While there was no strict necessity to switch from the fluorescence-based system used in *E. coli* to a luminescence-based system in A. tumefaciens, we opted for luminescence based on prior experience with other Alphaproteobacteria (e.g., 10.1128/mSystems.00893-21), where luminescence offered significant advantages. Specifically, it provides essentially background-free signal detection and greater sensitivity for monitoring gene expression. In addition, as demonstrated in Suppl. Fig. 9c and d, the luminescence system enables real-time tracking of gene expression dynamics, which further supported its use in our experimental setup (see our response to reviewer #2’s general comments).

(17) This is a very minor comment that the authors can take or leave, but I got hung up on the word "implement" when it appeared a few times in the manuscript because I tended to read it as "put a plan into place" rather than its other meaning.

In the abstract, we have replaced one instance of the word ‘implement’ by ‘instrument’.

(18) The authors should include the relevant constructs on AddGene or another public strainsharing service.

We whole-heartedly subscribe to the idea of freely sharing research materials with fellow scientists. Therefore, we had already deposited the most relevant AvNIRusk in Addgene, even prior to the initial submission of the manuscript (accession number 235084). In the meantime, we have released the deposition, and the plasmid can be obtained from Addgene since May 15_th_ of this year.

**Reviewer #3 (Recommendations for the authors):**
Suggestion for improvement:This paper relies heavily on variations in linker sequences to shift responses. I am familiar with prior work from the Moglich lab in which helical linkers were employed to shift responses in synthetic two-component systems, with interesting periodicity in responses with every 7 residues (as expected for an alpha helix) and inversion of responses at smaller linker shifts. There is no mention in this paper whether their current engineering follows a similar rationale, what types of linkers are employed (e.g. flexible vs helical), and whether there is an interpretation for how linker lengths alter responses. Can you explain what classes of linker sequences are used throughout Figures 2 and 3, and whether length or periodicity affects the outcome? This would be very helpful for readers who are new to this approach, or if the rationale here differs from the authors' prior work.

The PATCHY approach employed at present followed a closely similar rationale as in our previous studies. That is, linkers were extended/shortened and varied in their sequence by recombining different fragments of the natural linkers of the parental receptors, i.e., the bacteriophytochrome and the FixL sensor histidine kinase, respectively. We have added a statement to this effect in the text and a reference to Suppl. Fig. 3 which illustrates the principal approach.

Compared to our earlier studies, we isolated fewer receptor variants supporting light-regulated responses, despite covering a larger sequence space. Owing to the sparsity of the light-regulated variants, an interpretation of the linker properties and their correlation with light-regulated activity is challenging. Although doubtless unsatisfying from a mechanistic viewpoint, we therefore refrain from a pertinent discussion which would be premature and speculative at this point. As the reviewer raises a valid and important point, we have expanded the text by referring to our earlier studies and the observed dependence of functional properties on linker composition.

It is sometimes difficult to intuit or rationalize the differences in red/IR sensitivity across closely related variants. An important example appears in Figure 3C vs 3B. I think the AvTod+21 in 3B should be the equivalent to the DsRed response in the second column of 3C (AvTod+21 + DmDERusk), except, of course, that the bacteria in 3C carry an additional plasmid for the DERusk system. However, in 3B, the response to red light is substantial - ~50% as strong as that for IR, whereas in 3C, red light elicits no response at all. What is the difference? The reason this is important is that the AvTod+21 and DMDERusk represent the best "orthogonal" red and infrared light responses, but this is not at all obvious from 3B, where AvTod+21 still causes a substantial (and for orthogonality, undesirable) response under red light. Perhaps subtle differences in expression level due to plasmid changes cause these differences in light responses? Could the authors test how the expression level affects these responses? The paper would be greatly improved if observations of the diverse red/IR responses could be rationalized by some design criteria.

As noted above in our response to reviewer #2, we have now normalized all fluorescence readings to joint reference values, thus allowing a better comparison across experiments.

The reviewer is correct in noting that upon multiplexing, the individual plasmid systems support lower fluorescence levels than when used in isolation. We speculate that the combination of two plasmids may affect their copy numbers (despite the use of different resistance markers and origins of replications) and hence their performance. Likewise, the cellular metabolism may be affected when multiple plasmids are combined. These aspects may well account for the absent red-light response in AvTod+21 in the multiplexing experiments which is – indeed – unexpected. As, at present, we cannot provide a clear rationalization for this effect, we recommend verifying the performance of the plasmid setups when multiplexing.

The paper uses "red" and "infrared" to refer to ~624 nm and ~800 nm light, respectively. I wonder whether it might be possible to shift these peak wavelengths to obtain even better separation for the multiplexing experiments. Perhaps shifting the specific red wavelength could result in better separation between DERusk and AvTod systems, for example? Could the authors comment on this (maybe based on action spectra of their previously developed tools) or perhaps test a few additional stimulation wavelengths?

The choice of illumination wavelengths used in these experiments is dictated by the LED setups available for illumination of microtiter plates. On the one hand, we are using an SMD (surface-mount device) three-color LED with a fixed wavelength of the red channel around 624 nm (see Hennemann et al., 2018). On the other hand, we are deploying a custom-built device with LEDs emitting at around 800 nm (see Stüven et al., 2019 and this work). Adjusting these wavelengths is therefore challenging, although without doubt potentially interesting.

To address this reviewer comment, we have added a statement to the text that the excitation wavelengths may be varied to improve multiplexed applications.

Additional minor comments:(1) Figure 2C: It would be very helpful to place a legend on the figure panel for what the colors indicate, since they are unique to this panel and non-intuitive.

This comment coincides with one by reviewer #2, and we have added pertinent legends to this and related supplementary figures.

(2) Figure 3C: it is not obvious which system uses DsRed and which uses YPet in each combination, since the text indicates that all combinations were cloned, and this is not clearly described in the legend. Is it always the first construct in the figure legend listed for DsRed and the second for YPet?

For clarification, we have revised the x-axis labels in Fig. 3C. (And yes, it is as this reviewer surmises: the first of the two constructs harbored DsRed and the second one YPet.)